# Early detection of occupational stress: Enhancing workplace safety with machine learning and large language models

**Mohammad Junayed Hasan**[ID], **Jannat Sultana, Silvia Ahmed, Sifat Momen**[ID]*

Department of Electrical and Computer Engineering, North South University, Dhaka, Bangladesh

* sifat.momen@northsouth.edu

**Data availability statement:** The data used in this study can be accessed from https://data.mendeley.com/datasets/cgyh5s88kc/3 DOI: 10.17632/cgyh5s88kc.3 Synthetic data

## Abstract

Occupational stress is a major concern for employers and organizations as it compromises decision-making and overall safety of workers. Studies indicate that work-stress contributes to severe mental strain, increased accident rates, and in extreme cases, even suicides. This study aims to enhance early detection of occupational stress through machine learning (ML) methods, providing stakeholders with better insights into the underlying causes of stress to improve occupational safety. Utilizing a newly published workplace survey dataset, we developed a novel feature selection pipeline identifying 39 key indicators of work-stress. An ensemble of three ML models achieved a state-of-the-art accuracy of 90.32%, surpassing existing studies. The framework's generalizability was confirmed through a three-step validation technique: holdout-validation, 10-fold cross-validation, and external-validation with synthetic data generation, achieving an accuracy of 89% on unseen data. We also introduced a 1D-CNN to enable hierarchical and temporal learning from the data. Additionally, we created an algorithm to convert tabular data into texts with 100% information retention, facilitating domain analysis with large language models, revealing that occupational stress is more closely related to the biomedical domain than clinical or generalist domains. Ablation studies reinforced our feature selection pipeline, and revealed sociodemographic features as the most important. Explainable AI techniques identified excessive workload and ambiguity (27%), poor communication (17%), and a positive work environment (16%) as key stress factors. Unlike previous studies relying on clinical settings or biomarkers, our approach streamlines stress detection from simple survey questions, offering a real-time, deployable tool for periodic stress assessment in workplaces.

## Introduction

The well-being and safety of the working population is a crucial concern in today's modern society. Maintaining good health and ensuring workplace safety is crucial for individuals to lead productive and fulfilling lives, contributing to the overall prosperity of communities and nations. One key aspect of population health is occupational health [1–3], which refers to

generated in this study can be found from the supporting information files. Code can be found at https://github.com/junayed-hasan/occupational-stress-ml.

**Funding:** The author(s) received no specific funding for this work.

**Competing interests:** The authors have declared that no competing interests exist.

the physical, mental, and social well-being of individuals in relation to their work environments. Occupational stress [4,5], a prevalent workplace safety hazard in various professions, can have detrimental effects on individuals' health, impacting their productivity, job satisfaction, overall quality of life, and most critically, their workplace safety behavior. Studies have shown that stressed workers are significantly more likely to be involved in workplace accidents due to decreased attention, impaired decision-making, and compromised safety protocol adherence [6,7]. Literatures also indicate that prolonged exposure to heavy stress reduces life expectancy by 2.8 years [8]. For middle-aged individuals, the work-related stress is found to reduce healthy life expectancy by an average of 1.7 years [9], highlighting the critical need for effective stress detection and management systems in workplace safety protocols.

Occupational stress not only poses significant health and safety concerns for individuals but also creates substantial economic challenges for organizations and society as a whole. Studies, including [8], have highlighted the considerable costs related to absenteeism, presenteeism, and healthcare expenses stems from occupational stressors. From a workplace safety perspective, stressed employees are more likely to make errors, violate safety protocols, and be involved in workplace accidents, making stress detection and management a critical component of comprehensive safety systems. Addressing and mitigating occupational stress is therefore crucial, as it affects both individual safety and has broader implications for organizational effectiveness and societal well-being. Research, such as [10], indicates that higher stress levels often lead to increased absenteeism among employees – thus absenteeism can be an indicator of occupational stress. In their study, they employed machine learning techniques to predict absenteeism in a company – thus creating a model that aims to comprehend occupational stress.

Existing approaches to occupational stress detection have primarily relied on traditional methods such as self-report questionnaires [11,12], longitudinal studies [13,14], statistical analyses [15,16], and observational studies [17,18]. While these methods have contributed to our understanding of occupational stress, they present several critical limitations from a workplace safety perspective: (1) they often lack the ability to capture the complex interplay of factors that influence stress levels in diverse work environments, potentially missing early warning signs of safety risks; (2) their reactive nature fails to predict potential stress-related safety incidents before they occur; (3) the time-consuming and subjective nature of these approaches makes them impractical for real-time safety monitoring; and (4) they often operate in isolation, missing the potential synergies between different analytical approaches that could provide more comprehensive insights.

Furthermore, existing computational approaches to stress detection typically focus on either machine learning techniques or natural language processing in isolation, failing to leverage the complementary strengths of multiple AI domains. This siloed approach limits the accuracy and reliability of stress detection systems, with current state-of-the-art methods achieving accuracy rates below 85% in real-world applications. Additionally, most existing studies utilize older datasets that may not reflect current workplace dynamics and stressors, particularly in the context of evolving work environments and safety requirements. A particular gap exists in the integration of explainable AI techniques with stress detection systems, making it difficult for safety managers to understand and act upon the model's predictions.

The pressing need for more effective occupational stress detection and management is further emphasized by current workplace safety statistics. Stress-related safety incidents account for a significant portion of workplace accidents [6,7,19,20], yet existing detection methods lack the predictive capabilities necessary for proactive intervention. Consequently, there is an urgent need to explore innovative solutions that can effectively address these limitations and provide more comprehensive and proactive strategies for occupational stress management,

with the specific aims of: (1) achieving detection accuracy above 90% through multi-model integration; (2) providing real-time, interpretable predictions suitable for safety management applications; and (3) developing a scalable framework that can adapt to different workplace contexts while maintaining consistent performance metrics.

In this study, we comprehensively detect and analyze occupational stress by leveraging all three crucial domains of AI: machine learning [21], deep learning [22–24], and natural language processing [25,26]. Our work utilizes a newly published dataset from February 2024, consisting of Malaysian workplace survey data that has not been previously used for occupational stress detection. This temporal relevance ensures our findings reflect current workplace dynamics and safety challenges. The methodology introduces several innovative components: (1) a systematic preprocessing pipeline that handles complex survey responses while preserving their safety implications; (2) a novel feature fusion approach that combines multiple feature selection techniques, demonstrated to improve model performance by 5-10% compared to single-technique approaches; and (3) a unique algorithm for converting tabular data into natural language sentences, enabling more nuanced domain analysis [25] of stress factors through large language models (LLMs).

The preprocessed data is used to train and evaluate 11 different machine learning algorithms, a one-dimensional deep convolutional neural network (1D-CNN), and five LLMs for stress classification. Domain analysis with LLMs revealed that occupational stress patterns align more closely with biomedical domains than clinical domains, with biomedical language models outperforming their clinical counterparts in stress detection. The model's generalizability was further validated through extensive synthetic data experiments, demonstrating robust performance across different workplace scenarios and strengthening its reliability for real-world safety applications. The main contributions of this study are summarized below:

- A pioneering integration of machine learning, deep learning, and natural language processing for occupational stress detection, achieving a previously unattained accuracy of 90.32% in stress classification.
- The first application of AI techniques to a novel Malaysian survey dataset, providing fresh insights into contemporary workplace stress patterns through detailed analysis with ablation studies and explainable-AI.
- Empirical demonstration that combining feature importance and feature ranking techniques significantly enhances model performance by 5-10%, extracting more predictive features than individual methods.
- Development of a novel algorithm for converting tabular occupational stress data into natural language sentences, enabling more nuanced stress analysis through LLMs with a 100% information preservation rate.
- Superior performance over eight recent state-of-the-art methods on the same dataset, with the ensemble model achieving the highest recorded accuracy (90.32%) and F1-score (89.20%) in occupational stress detection.
- Deployment of the developed model on a public repository, making it immediately accessible for practical workplace safety applications, with documented response times under 100ms for real-time stress assessment.
- Extensive validation of model generalizability through synthetic data generation using four different techniques (Gaussian Copula, CTGAN, TVAE, and Copula GAN), achieving 89.00% accuracy on unseen test scenarios and demonstrating robust performance in varied workplace contexts.

These contributions collectively advance the field of occupational safety science by providing a more accurate, interpretable, and practical approach to stress detection and management. The framework's ability to process diverse data types and provide explainable results makes it particularly valuable for safety managers and organizational decision-makers. Our quantitative improvements over existing methods, combined with the framework's practical deployability, represent a significant step forward in proactive workplace safety management through stress detection and mitigation.

The remainder of this article is structured as follows: The *Related work* section reviews relevant literature and identifies current research gaps. The *Materials and Methods* section describes the dataset, exploratory data analysis, computational methods, ensemble modeling approach, natural language sentence generation algorithm, and synthetic data generation techniques. The *Results* section presents experimental findings, including comparisons with existing methods, external validation using synthetic data, ablation studies, and explainable AI analyses of safety-critical features. The *Discussion* section explores the methodological significance of the study, its implications for workplace safety management, and its limitations, along with directions for future research. Finally, the *Conclusions* section summarizes the main findings, contributions, limitations, and potential future directions.

## Related work

Occupational stress has been extensively studied due to its significant implications for workplace safety, individual well-being, and organizational performance. The literature can be categorized into three main streams: traditional safety and health studies, historical perspectives, and computational approaches to stress detection and management.

In the context of workplace safety and health outcomes, various studies have established critical links between occupational stress and serious health risks. Research has demonstrated strong correlations between job stress and cardiovascular health [27,28], metabolic syndrome [29], and other safety-critical health outcomes. Organizational safety factors, including workplace dynamics and social support systems, have been shown to significantly influence stress levels and job satisfaction [4,30–32], directly impacting workplace safety behaviors.

The role of organizational safety climate in stress management has been extensively investigated. Studies have highlighted how job control, work demands, and social support systems can mitigate stress-related safety risks [33–35]. Work-family conflict and organizational justice have emerged as significant factors affecting workplace safety culture [36]. Physical safety impacts of occupational stress have been documented across diverse professional contexts [37–40], emphasizing the need for comprehensive stress management in workplace safety protocols.

From a historical perspective, the recognition of occupational stress as a workplace safety concern dates back to ancient times [41]. Cross-cultural studies have revealed that stress affects workplace safety and mental health differently across various cultural contexts [42], with psychological factors like self-efficacy serving as protective mechanisms [43]. This historical understanding has shaped modern approaches to workplace safety management.

A growing body of AI-based research has recently emerged to address occupational stress detection more effectively. For instance, [44] applied four machine learning classifiers (random forest, support vector machine, K-nearest neighbors, and artificial neural network) and compared them with logistic regression to predict chronic stress in medical practice assistants. The random forest classifier yielded the best performance, improving the area under the curve by over 20% compared to the logistic model. The authors identified excessive workload, high demand for concentration, and insufficient leadership support as key contributors to stress.

Similarly, [45] investigated stress prediction using self-reported data and biomarkers, highlighting that wearable technologies combined with machine learning can reveal new insights into employees' stress patterns.

[46] proposed a stress prediction method using the Perceived Stress Scale and machine learning, where logistic regression provided a 99% accuracy, underscoring the relevance of questionnaire-based data for reliable stress detection. In the context of the COVID-19 pandemic, [47] utilized the XGBoost algorithm to predict employee stress, finding that working hours, workload, age, and role ambiguity significantly influenced performance. Focusing on anxiety state detection, [48] demonstrated that XGBoost performed best, and further employed SHAP to interpret their model. In [49], a hybrid depression assessment scale was developed, and multiple machine learning and deep learning models were tested. Random Forest achieved the highest accuracy of 98.08%, with LIME explanations providing transparent insights into model decisions. Elsewhere, [50] explored machine learning approaches for predicting depression risk in workplaces, finding that random forest had the highest accuracy (88.7%) and revealing gender, physical health, and psychosocial risk/protective factors as critical influences. Lastly, [26] extended these AI methodologies to the domain of life satisfaction, converting tabular data into natural language for large language model processing and reaching an accuracy of 93.80%. The authors highlighted the importance of interpretability and domain adaptation in model deployment.

Table 1 provides a comprehensive summary of key studies in occupational stress detection, highlighting diverse methodologies, findings, and limitations across various domains and safety contexts.

These studies capture the complex relationship between occupational stress and workplace safety. While the continuous development of AI techniques offers promising avenues for stress detection and management, existing approaches often lack integration between different methodological domains and fail to provide real-time, interpretable results suitable for practical safety management applications. Additionally, most current studies utilize historical datasets that may not reflect contemporary workplace safety challenges and stressors. This gap highlights the need for innovative, integrated approaches that can leverage multiple AI domains while maintaining interpretability for safety management applications.

## Materials and methods

### Materials

**Dataset.** The dataset utilized in this study is obtained from a recently published research article by Majid et al. [52], released in February 2024. The data were collected from 11 November 2021 until 30 October 2022, focusing primarily on occupational stress, workplace safety indicators, job satisfaction, and job performance among Malaysian workers. The study employed a quantitative research approach through comprehensive questionnaire development and survey methodology. A sample of 309 participants from diverse occupational backgrounds was selected using simple random sampling, representing various workplace environments and safety contexts. The questionnaire gathered extensive information on respondents' demographics, occupational stress factors, safety behaviors, job satisfaction metrics, and performance indicators. Ethical considerations were addressed through proper informed consent procedures. Additionally, the dataset does not contain any missing values. This contemporary dataset provides valuable insights into the complex relationships between occupational factors and workplace safety, particularly in the context of organizational health maintenance and stress management.

**Table 1. Summary of the relevant studies in occupational stress detection.**

| Study | Domain/Industry | Methodology | Key Findings | Limitations |
|---|---|---|---|---|
| *Traditional Approaches* | | | | |
| [27] | Healthcare | Effort-reward imbalance model and job strain model | Imbalance between efforts and rewards significantly increased risk of coronary heart disease | Job strain model was less predictive |
| [28] | Steel Industry | Prospective cohort study | High job strain and effort-reward imbalance elevated cardiovascular mortality risk | Results specific to Finnish metal industry may not generalize |
| [29] | Various Professions | Longitudinal study | Chronic work stress more than doubled the risk of metabolic syndrome | Limited to specific professions |
| [30] | Organizational | Survey and observational study | Negative workplace dynamics are associated with withdrawal behaviors, job dissatisfaction, and burnout | Subjective measures |
| [31] | Healthcare | Survey | High levels of burnout and psychiatric morbidity among UK consultants | Specific to UK consultants |
| [4] | Healthcare | Survey | Occupational stress in nurses linked to declines in job performance | Focused on nurses |
| [32] | Various Professions | Meta-analysis | Social support buffers the negative effects of stress | Conflicting results |
| [33] | Education | Survey | Social support reduces burnout, especially with positive feedback from supervisors | Context-specific |
| [34] | Various Professions | Survey | High job demands and low decision latitude lead to higher strain and burnout | Limited to specific contexts |
| [35] | Various Professions | Survey | High work demands and low social support linked to disturbed sleep and stress | Limited to specific contexts |
| [36] | Education | Survey | Work-family conflict mediates relationship between organizational justice and stress | Specific to university faculty |
| [37] | Accounting | Historical analysis | Cyclic occupational stress showed increased serum cholesterol and accelerated blood clotting times | Specific to accounting profession |
| [38] | Various Professions | Survey and observational study | Mobbing has severe mental and psychosomatic health consequences comparable to PTSD | Subjective measures |
| [39] | Various Professions | Survey | Bullying at work leads to lower social support and increased symptoms of anxiety | Specific to bullying contexts |
| [40] | Various Professions | Meta-analysis | High job strain and low job control associated with increased blood pressure | Mixed findings |
| *Historical and Comparative Studies* | | | | |
| [41] | Historical | Literature review | Historical perspectives on occupational stress highlight longstanding recognition | General overview |
| [42] | Various Professions | Comparative study | Occupational stress affects mental health differently across cultures | Culture-specific |
| [43] | Education | Survey | Self-efficacy protects against job strain and burnout, especially for younger teachers | Focus on teachers |
| *Machine Learning Approaches* | | | | |
| [44] | Tech Industry | Machine learning classifiers | Random forest and support vector machines predicted chronic stress with high accuracy | Traditional logistic regression models underperformed |
| [45] | Healthcare | Wearable devices and biomarkers | Wearable devices combined with machine learning offer new avenues for monitoring stress | Dependent on quality and availability of wearable devices |
| [46] | Healthcare | Perceived Stress Scale technique | Logistic regression and random forest models showed high accuracy in stress prediction | Limited to perceived stress and does not account for objective measures |
| [47] | Tech Industry | Machine learning models for predicting stress levels | Identified working hours and role ambiguity as significant predictors of stress | Focus on pandemic-specific factors may limit applicability to other contexts |
| [48] | Tech Industry | Machine learning techniques for stress prediction | Identified significant predictors of stress during COVID-19 pandemic | Dependent on data quality and specific healthcare settings |
| [51] | Steel Industry | Machine learning algorithms for stress prediction | Predicted stress levels in insurance employees with high accuracy | Data limitations and industry-specific factors may affect generalizability |
| [49] | Education | Machine learning models for predicting stress levels | Identified working hours and role ambiguity as significant predictors of stress | Model performance may vary across different industries |
| [50] | Various Professions | Machine learning models | Predicted stress levels with high accuracy using socioeconomic data during pandemic | Focus on pandemic-specific factors |
| [26] | Various Professions | Machine learning and large language models | Predicting life satisfaction and well-being using converted tabular data | Specific to Danish population |

**Exploratory data analysis.** The dataset is structured into four primary sections relevant to workplace safety and stress management: sociodemographic information (Section A), occupational stress indicators (Section B), job satisfaction metrics (Section C), and job performance measures (Section D). An additional health-related section (Section E) was also included in the original dataset. The responses are categorized into: (i) Nominal categorical variables comprising sociodemographic information, and (ii) Ordinal survey responses about occupational stress, job satisfaction, and job performance on a Likert scale, providing a comprehensive view of workplace safety dynamics.

Fig 1 illustrates the distribution of sociodemographic factors that could influence workplace stress and safety behaviors. Key safety-relevant observations include: a majority (67.3%) of respondents are in their prime working years (30-39 years), with most (91.8%) holding a bachelor's degree or lower qualifications. Notably, 97.1% are employed full-time, and 62.8% have over 10 years of working experience, suggesting significant exposure to workplace stressors. The household income distribution follows a normal curve, with most participants (39.8%) in the middle-income bracket (RM3,970-RM7,099), which could influence workplace stress levels and safety behaviors.

The dataset includes 41 survey questions pertaining to occupational stress, labeled OS1 to OS41. These questions cover various factors influencing occupational stress, including workload demands, control over work, support from managers and peers, job role clarity, organizational changes, interpersonal relationships, and work-life balance. The survey employs a Likert scale to measure these dimensions.

Fig 2a presents grouped box plots showing the distribution of stress levels across different occupational safety categories. A higher value (5) indicates better stress management, while a lower value (1) suggests potentially hazardous stress levels. The analysis reveals that workload demands and work-family conflict are the primary contributors to occupational stress,

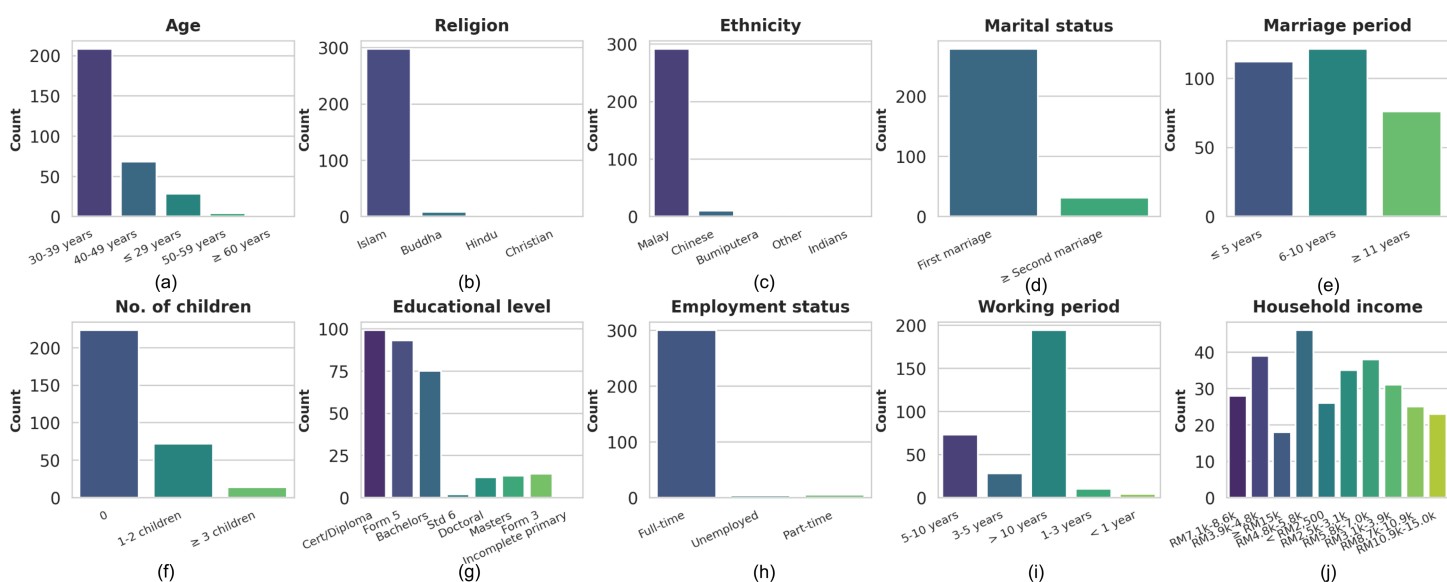

**Fig 1. Demographic and Socioeconomic Characteristics of the Survey Respondents: (a) Age Group Distribution, (b) Religion Distribution, (c) Ethnicity Distribution, (d) Marital Status Distribution, (e) Marriage Period Distribution, (f) Number of Children Distribution, (g) Educational Level Distribution, (h) Employment Status Distribution, (i) Working Period Distribution, and (j) Household Income Distribution.**

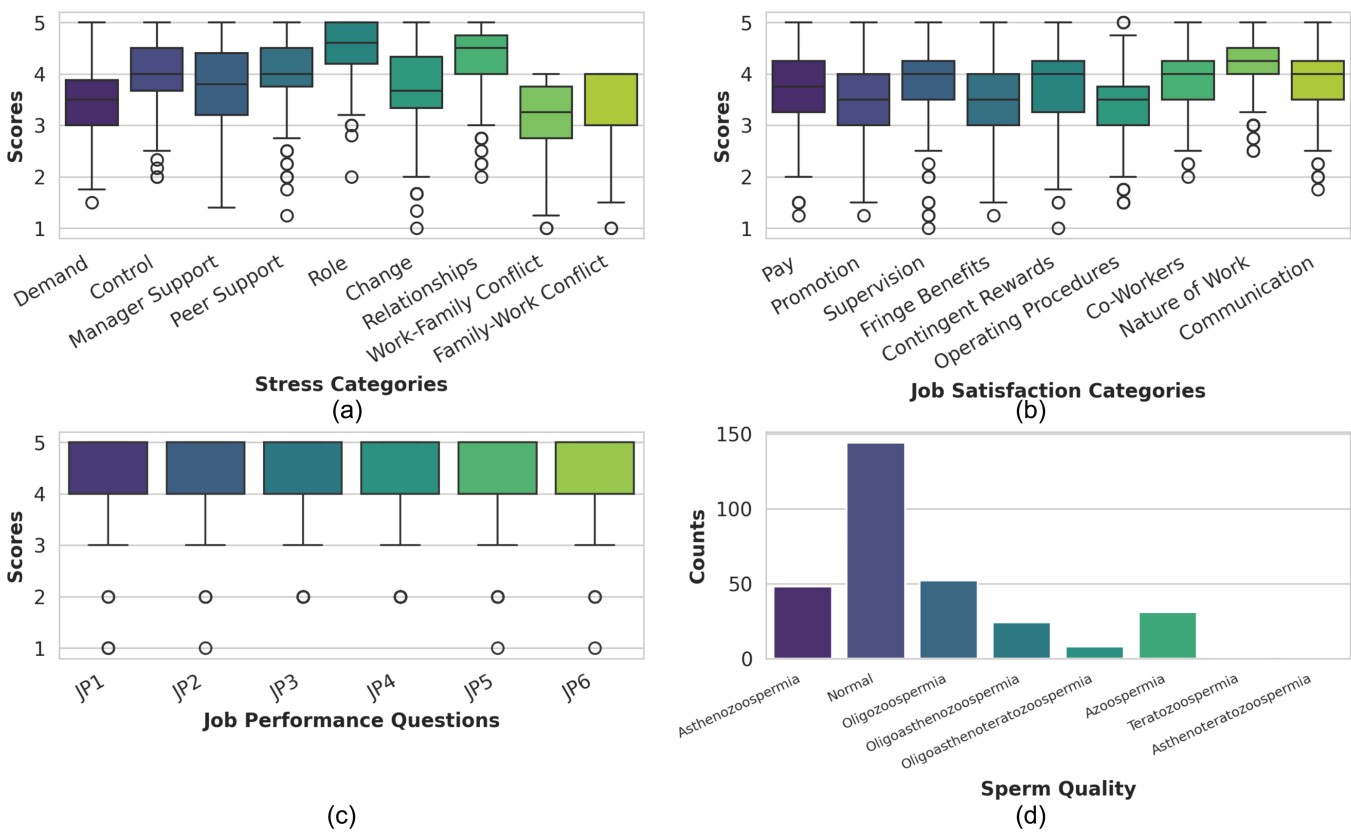

**Fig 2. Various data visualizations from the study: (a) Grouped box plots showing the data distribution for each occupational stress category, (b) Grouped box plots showing the data distribution for each job satisfaction category, (c) Box plots showing the data distribution for the responses of each job performance survey question, and (d) Bar plot showing the distribution of sperm quality among the respondents of the dataset used in the study.**

potentially compromising workplace safety. The uniformity in manager support scores across respondents suggests its consistent role in workplace stress management.

The job satisfaction component comprises 36 survey questions (JS1-JS36), categorized into dimensions critical for workplace safety: pay adequacy, career progression, supervisory support, organizational benefits, reward systems, operational procedures, team dynamics, work nature, and safety communication. As shown in Fig 2b, satisfaction levels vary across categories, with operational procedures and promotion opportunities showing the lowest satisfaction scores, potentially indicating areas of stress that could impact safety behavior.

Job performance metrics (JP1-JP6) focus on two safety-critical aspects: work completion efficiency (JP1-JP3) and distraction avoidance (JP4-JP6). Fig 2c demonstrates consistent response patterns across these metrics, indicating reliable measurement of performance factors that could affect workplace safety. Additionally, all six items use a five point Likert scale (values from 1–5), with respondents clustering around similar score ranges for all questions. Since these items capture closely related facets of job performance under fertility challenges, their responses exhibit high correlation, resulting in overlapping medians, quartiles, and whiskers in the box plots. While most respondents report high performance levels, the presence of outliers suggests varying degrees of stress impact on work execution.

Lastly, the distribution of the sperm quality of the respondents is shown in Fig 2d. The bar plot shows that most of the respondents have a normal sperm quality type. However, a good

portion of the respondents suffer from Oligozoospermia and Asthenozoospermia. The number of respondents suffering from Teratozoospermia and Asthenoteratozoospermia is the least, which are close to zero (one for each).

This comprehensive dataset enables detailed analysis of the relationships between occupational stress, workplace safety, and organizational performance, providing a solid foundation for developing predictive models for stress detection and management in workplace safety contexts.

**AI models.** We employed a diverse set of ML algorithms, including Random Forest, AdaBoost, Decision Tree, Logistic Regression, Support Vector Classifier (SVC), K-Nearest Neighbors (KNN), Gaussian Naive Bayes, XGBoost, and LightGBM. Additionally, we developed a customized one-dimensional convolutional neural network (1D CNN) [22,24,53] to capture temporal patterns and dependencies in the survey response data, as illustrated in Fig 3. This 1D CNN architecture comprises three convolutional layers with batch normalization, ReLU activation, and max pooling, followed by two fully connected layers.

Let $\mathbf{X} \in \mathbb{R}^{B \times 1 \times L}$ represent a batch of $B$ survey instances, each viewed as a single-channel sequence of length $L$ (e.g., after encoding 39 features or time steps). Our 1D CNN (Fig 3) applies a series of convolutional, batch normalization, ReLU, and max pooling operations to extract hierarchical patterns, followed by fully connected layers for classification. Concretely, at the $i$-th convolutional layer with weights $\mathbf{W}^{(i)}$ and biases $\mathbf{b}^{(i)}$, the output $\mathbf{Z}^{(i)}$ is computed as

$$\mathbf{Z}^{(i)} = \mathrm{MaxPool}\Big(\mathrm{ReLU}\big(\mathrm{BatchNorm}(\mathbf{W}^{(i)} \star \mathbf{Z}^{(i-1)} + \mathbf{b}^{(i)})\big)\Big) \tag{1}$$

where $\star$ denotes the 1D convolution operator, $\mathrm{ReLU}(\mathbf{x}) = \max(\mathbf{0}, \mathbf{x})$, and BatchNorm normalizes feature maps to stabilize training. The first convolutional layer takes $\mathbf{Z}^{(0)} = \mathbf{X}$, while subsequent layers take $\mathbf{Z}^{(i-1)}$ from the previous layer. After the third convolutional block, the feature maps are flattened and passed to fully connected layers:

$$\mathbf{h}_1 = \mathrm{ReLU}\big(\mathbf{W}_{fc1}\,\mathrm{Flatten}(\mathbf{Z}^{(3)}) + \mathbf{b}_{fc1}\big), \quad \mathbf{h}_2 = \mathrm{Dropout}(0.5)\big(\mathbf{h}_1\big), \tag{2}$$

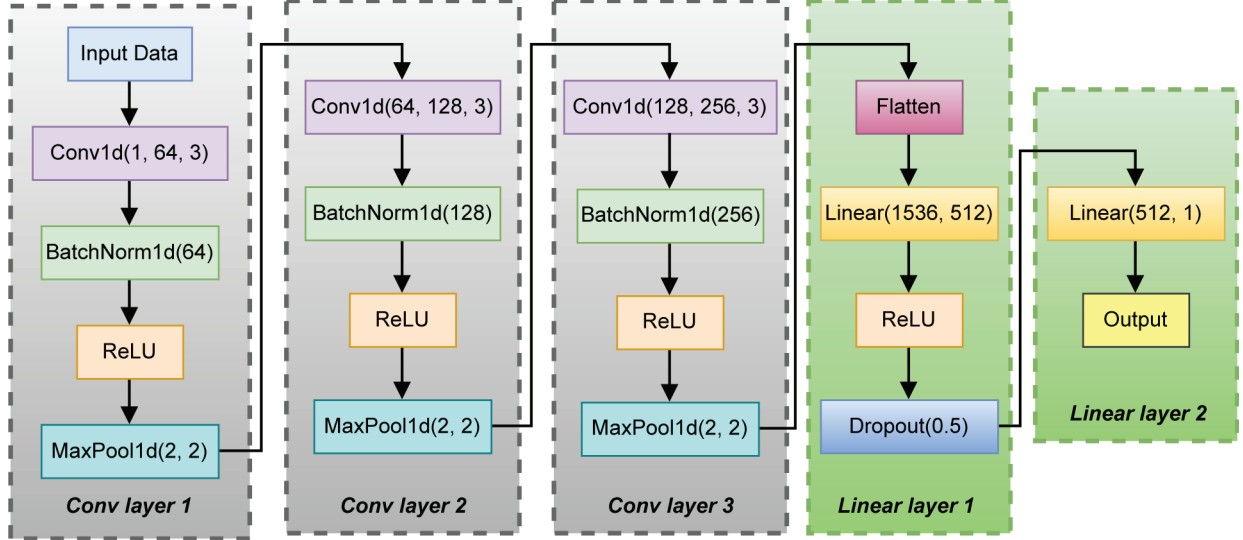

**Fig 3. Architecture of the proposed 1D CNN for occupational stress detection.**

then

$$y = \sigma(\mathbf{W}_{fc2}\,\mathbf{h}_2 + \mathbf{b}_{fc2}), \tag{3}$$

where $\sigma(\cdot)$ is the sigmoid function producing a scalar output $y \in (0, 1)$ denoting the probability of "no stress" (or "stressed," depending on labeling). The network's trainable parameters $\{\mathbf{W}^{(i)}, \mathbf{b}^{(i)}, \mathbf{W}_{fcj}, \mathbf{b}_{fcj}\}$ are optimized via the binary cross-entropy loss:

$$\mathcal{L}(\theta) = -\frac{1}{B}\sum_{n=1}^{B}\Big[t_n\,\log y_n + (1 - t_n)\,\log(1 - y_n)\Big], \tag{4}$$

where $t_n \in \{0, 1\}$ is the ground truth label for sample $n$ and $y_n$ is the predicted probability. The resulting network has 911,873 trainable parameters, computed by summing contributions from each layer:

- Conv1d(1, 64, 3): $(1 \times 64 \times 3) + 64 = 192 + 64 = 256$
  BatchNorm1d(64): $2 \times 64 = 128$
  Total: $256 + 128 = 384$
- Conv1d(64, 128, 3): $(64 \times 128 \times 3) + 128 = 24{,}576 + 128 = 24{,}704$
  BatchNorm1d(128): $2 \times 128 = 256$
  Total: $24{,}704 + 256 = 24{,}960$
- Conv1d(128, 256, 3): $(128 \times 256 \times 3) + 256 = 98{,}304 + 256 = 98{,}560$
  BatchNorm1d(256): $2 \times 256 = 512$
  Total: $98{,}560 + 512 = 99{,}072$
- Linear(1536, 512): $(1536 \times 512) + 512 = 786{,}432 + 512 = 786{,}944$
- Dropout(0.5): No trainable parameters.
- Linear(512, 1): $(512 \times 1) + 1 = 512 + 1 = 513$

Total trainable parameters: $384 + 24{,}960 + 99{,}072 + 786{,}944 + 513 = 911{,}873$

Through hierarchical feature extraction, the 1D CNN aims to effectively capture non-linear relationships among occupational stress factors and provide better performance and insights than many simpler statistical ML models that lack this capacity. Moreover, from computational perspective, it provides a balance between simpler models and more complex, large models that require no or millions of parameters, compared to its 911K parameters only.

To enable natural language processing and domain analysis [25] on occupational stress data, we also utilized LLMs from a diverse set of pre-training domains such as BERT [54] (general domain), BioBERT [55] (biomedical domain), ClinicalBERT [56] (clinical domain), DischargeBERT [57] (clinical domain), and COReBERT [58,59] (both biomedical and clinical domains) to classify generated natural language sentences for a comprehensive analysis of occupational stress, incorporating both clinical and biomedical contexts.

**Implementation details.** All experiments and inferences were conducted on an Amazon Linux AMI operating system, utilizing an NVIDIA T4 14GB Tensor Core GPU with 32GB of RAM on the Amazon AWS EC2 cloud server. Each experiment was performed five times with the following random seeds: 1, 13, 24, 37, and 42. Scikit-learn [60], numpy [61], pandas [62], matplotlib [63], seaborn [64], and scipy [62] were used for ML algorithms, data manipulation, visualization, and statistical analyses, ensuring robust and reproducible results. The 1D CNN was implemented using PyTorch [65]. The Binary Cross Entropy loss function and the Adam optimizer were used [66]. For the LLMs, PyTorch and the Transformers library [67] were used. Texts were tokenized with WordPiece tokenizers, processing the first 512 tokens due to BERT-like model limitations. Each experiment was repeated three times, with results reported

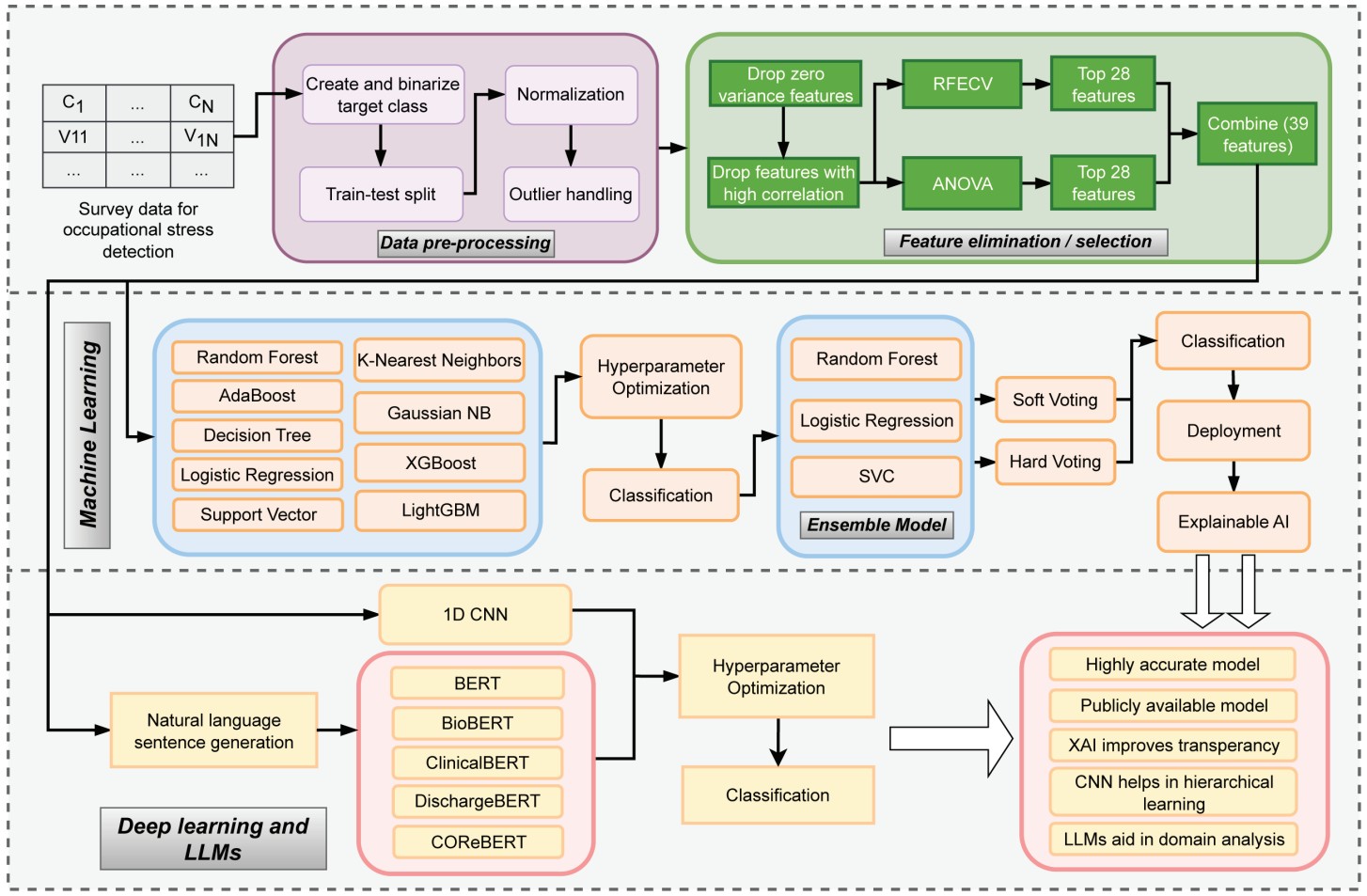

**Fig 4. Schematic representation of the end-to-end occupational stress detection system architecture, illustrating the workflow from data acquisition through pre-processing, feature selection, and classification using ML algorithms, 1D CNN, and LLMs. The diagram highlights key steps including data pre-processing, feature elimination and selection techniques, natural language sentence generation, hyperparameter optimization, and the creation of an ensemble model, culminating in a deployable and explainable AI-driven stress detection system.**

as mean and standard deviation. The AdamW optimizer [68] was employed used. The performance of the models was evaluated using accuracy, precision, recall, macro-averaged F1 score, and ROC-AUC score. The codes, implementation details and notebooks supporting the findings of this study is publicly available at https://github.com/junayed-hasan/occupational-stress-ml/.

## Methods

**System architecture and computational algorithm.** The system architecture for this study, shown in Fig 4, begins with survey data acquisition for occupational stress detection, followed by data pre-processing, feature selection, and classification using machine learning algorithms, a 1D CNN, and LLMs. An ensemble model is created with the three best-performing machine learning models, which is then deployed and explained using Explainable AI.

The overall computational approach is detailed in Algorithm 1. The algorithm consists of three main phases: (1) feature selection through RFECV-ANOVA integration, (2) natural

**Algorithm 1.   Overall computational algorithm of the proposed framework.**

**Input:**
- $D$: Input dataset with $n$ samples and $m$ features
- k: Number of cross-validation folds
- $t_{corr}$: Correlation threshold (0.8)
- $n_{feat}$: Target number of features (28)

**Output:**

- $F_{selected}$: Optimal feature set
- $S_{generated}$: Natural language sentences
- $M_{ensemble}$: Final ensemble model

**Phase 1: Feature Selection**

$F_{initial} \leftarrow$ PreprocessFeatures($D$) ;                    // Remove zero variance
$F_{uncorr} \leftarrow$ FilterCorrelated($F_{initial}, t_{corr}$);
Initialize $F_{RFECV} \leftarrow \varnothing$;
**foreach** $f \in F_{uncorr}$ **do**
  Train RandomForest with cross-validation;
  $importance_f \leftarrow$ GetFeatureImportance($f$);
  **if** $importance_f >$ *threshold* **then**
    $F_{RFECV} \leftarrow F_{RFECV} \cup \{f\}$;
  **end**
**end**
$F_{RFECV} \leftarrow$ SelectTop($F_{RFECV}, n_{feat}$);
Initialize $F_{ANOVA} \leftarrow \varnothing$;
**foreach** $f \in F_{uncorr}$ **do**
  $F$-score $\leftarrow$ ComputeANOVA($f$, target);
  $F_{ANOVA} \leftarrow F_{ANOVA} \cup \{(f, F\text{-score})\}$;
**end**
$F_{ANOVA} \leftarrow$ SelectTop($F_{ANOVA}, n_{feat}$);
$F_{selected} \leftarrow F_{RFECV} \cup F_{ANOVA}$ ;                    // Union of features
**end**

**Phase 2: Natural Language Generation**

Initialize $S_{generated} \leftarrow \varnothing$;
**foreach** *sample* $\in D$ **do**
  $text \leftarrow \varnothing$;
  $mapped\_values \leftarrow$ Map numerical to text responses;
  **foreach** $f \in F_{selected}$ **do**
    $context \leftarrow$ GetContext($f$);
    $value \leftarrow mapped\_values[f]$;
    $sentence \leftarrow$ FormatSentence($context, value$);
    $text \leftarrow text + sentence$;
  **end**
  $S_{generated} \leftarrow S_{generated} \cup$ Finalize($text$);
**end**
**end**

**Phase 3: Model Training and Ensemble Creation**

Train RandomForest on $F_{selected}$;
Train LogisticRegression on $F_{selected}$;
Train SVC on $F_{selected}$;
$M_{ensemble} \leftarrow$ CreateEnsemble(RF, LR, SVC);
Train 1D-CNN on $F_{selected}$;
Fine-tune BERT models on $S_{generated}$;
**end**
**return** $F_{selected}, S_{generated}, M_{ensemble}$;

language sentence generation, and (3) model training and ensemble creation. The algorithm's modular design ensures extensibility and adaptability to different workplace safety contexts while maintaining computational efficiency.

**Creating targets and binarizing target class.** The 41 survey questions related to occupational stress were aggregated to produce a single value representing the occupational stress level for each individual. The composite stress score $OS$ for an individual is calculated as:

$$OS = \sum_{i=1}^{41} OS_i \tag{5}$$

where $OS_i$ denotes the value of the $i$-th occupational stress survey question.

The aggregation process is illustrated in Fig 5a. The composite scores were normalized using Min-Max Scaling:

$$OS_{\text{normalized}} = \frac{OS - OS_{\min}}{OS_{\max} - OS_{\min}} \tag{6}$$

The normalized scores were then binarized based on a threshold of 0.5:

$$Y = \begin{cases} 0 & \text{if } OS_{\text{normalized}} < 0.5 \\ 1 & \text{if } OS_{\text{normalized}} \geq 0.5 \end{cases} \tag{7}$$

where $Y$ represents the binarized target class (0 indicates stress, 1 indicates no stress).

The class distribution after conversion is shown in Fig 5b, revealing imbalanced data with approximately 65% of samples in the "No Stress" class and 35% in the "Stress" class.

**Data preprocessing.** The dataset was divided into training and testing sets using an 80-20 split ratio. This process was performed five times with different random seeds, and the mean results were reported with standard deviations.

Data normalization was performed using Min-Max Scaling on both training and testing data, ensuring all feature values are within the same range. Outliers were identified and handled based on the Interquartile Range (IQR) method for the training set only. Data points were considered outliers if they fell outside the following bounds:

$$\text{Lower Bound} = Q1 - 1.5 \times \text{IQR} \tag{8}$$

$$\text{Upper Bound} = Q3 + 1.5 \times \text{IQR} \tag{9}$$

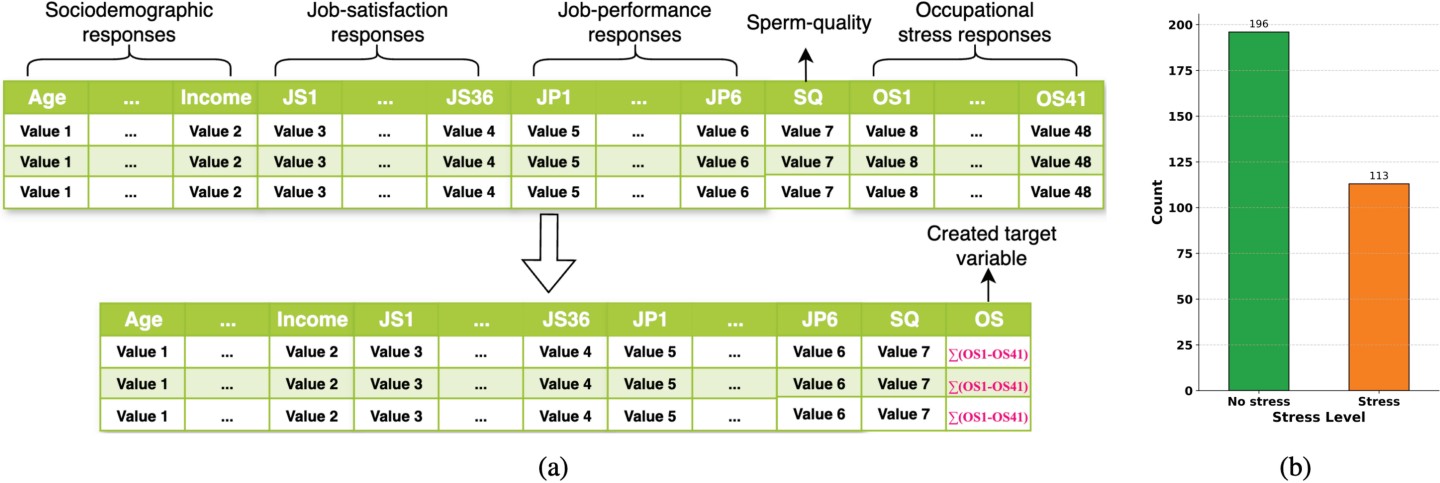

(a)

(b)

**Fig 5. (a) The aggregation process used to convert the 41 occupational stress columns into one single column representing occupational stress level of a person. (b) The class distribution of the data after creating the targets and binarizing the target class.**

Outliers were replaced by rounding the mean value of all other non-outlier data points.

**Feature elimination.** The feature elimination process involved removing features with zero variance and those exhibiting high correlation with other features. Zero variance features (Religion, Ethnicity, Marital Status, Employment Status, JS9, and JS27) were removed. Features with a Pearson's correlation coefficient greater than or equal to 0.8 were considered highly correlated, and one feature from each pair was removed based on domain knowledge and exploratory data analysis.

**Feature selection.** Feature selection was performed using a hybrid approach combining Recursive Feature Elimination with Cross-Validation (RFECV) and Analysis of Variance (ANOVA). RFECV's ability to capture complex non-linear relationships and feature interactions, and ANOVA's statistical power in identifying individually significant features was utilized to strengthen the approach.

The RFECV algorithm was implemented with a classifier as the base estimator, using 5-fold cross-validation and balanced accuracy as the scoring metric. Through this process, the optimal number of features was determined to be 28, based on the point where feature addition no longer significantly improved cross-validation scores. Concurrently, ANOVA was performed to rank features according to their F-values, which measure the ratio of between-group to within-group variance, with higher values indicating stronger discriminative power.

Rather than selecting features from either method alone, we employed a union-based integration strategy. The top 28 features from each method were identified and combined, resulting in 39 unique features after accounting for overlap. This integration yielded 11 additional features that would have been missed by using either method in isolation, demonstrating the complementary nature of the two approaches.

To validate our hybrid selection approach, we conducted ablation studies which are presented in the Results section in Table 6. The ablations included features selected by RFECV alone, features selected by ANOVA alone, and our integrated feature set. Results showed that models trained on the integrated feature set achieved significantly higher predictive performance. This empirical evidence supports our claim that the hybrid approach extracts more predictive features than individual methods.

Fig 6 shows the comparison of feature importances identified by RFECV and ANOVA F-values. The RFECV method identified JS12, JS35, and JS6 as the top three important features, while ANOVA identified JS6, JS12, and JS35 as the most significant features based on

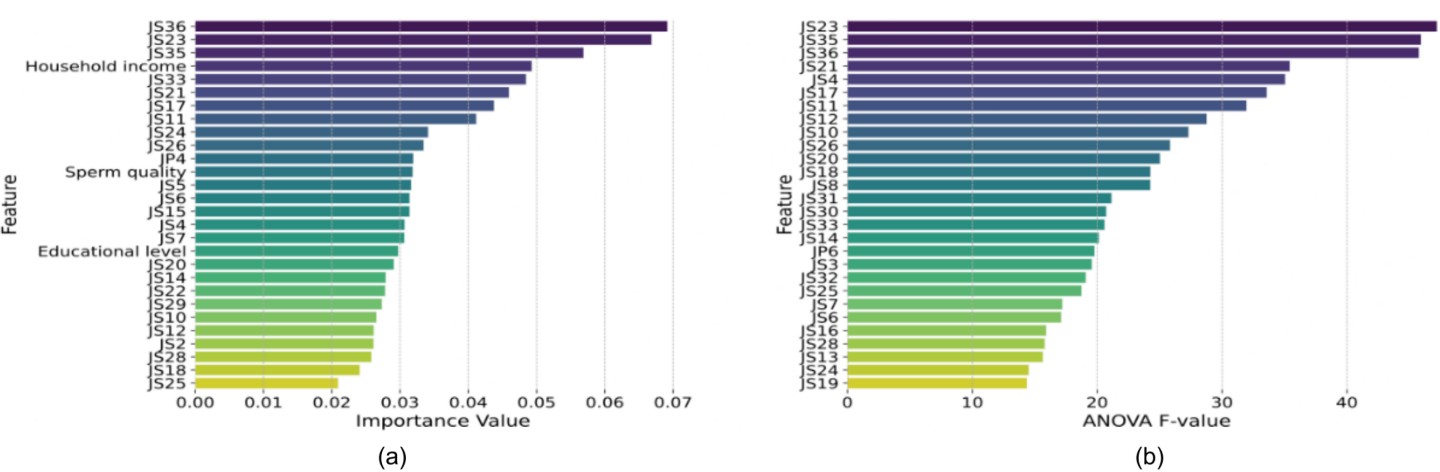

(a)                                        (b)

**Fig 6. Comparison of feature importances identified by (a) RFECV and (b) ANOVA F-values.**

F-values. Both methods consistently highlighted the importance of job satisfaction-related features in predicting occupational stress.

Fig 7 displays the 39 most important indicators of occupational stress extracted through this methodology. The top features include various aspects of job satisfaction (JS), job performance (JP), and demographic factors such as age group and working period. This comprehensive set of features provides a multifaceted view of the factors contributing to occupational stress.

**Hyperparameter optimization.** Hyperparameter optimization is a critical step in developing effective machine learning models. It involves finding the best set of hyperparameters that maximize model performance. Due to the high number of possible combinations, RandomizedSearchCV was employed instead of GridSearchCV to efficiently explore the hyperparameter space of the used machine learning, deep learning and large language models. Table 2 shows the models, selected hyperparameters, hyperparameter spaces explored for each model, and the selected hyperparameters of the models.

**Ensemble creation.** The ensemble learning involves combining multiple machine learning models to improve overall predictive performance. Ensembles leverage the strengths of individual models, reduce overfitting, and enhance generalization. In this study, an ensemble was created by selecting the three best-performing models: Random Forest Classifier, Logistic Regression, and Support Vector Classifier (SVC). The ensemble methods used were hard voting and soft voting.

Hard voting, also known as majority voting, involves taking the mode of the predicted classes from each individual model. Mathematically, for an ensemble of $n$ models, the hard voting prediction $\hat{y}$ for an instance $x$ can be represented as:

$$\hat{y} = \mathrm{mode}(\hat{y}_1, \hat{y}_2, \ldots, \hat{y}_n) \tag{10}$$

where $\hat{y}_i$ is the prediction from the $i$-th model.

Soft voting involves averaging the predicted probabilities from each individual model and selecting the class with the highest average probability. Mathematically, the soft voting prediction $\hat{y}$ for an instance $x$ is given by:

$$\hat{y} = \arg\max_c \left( \frac{1}{n} \sum_{i=1}^{n} P_i(y = c \mid x) \right) \tag{11}$$

where $P_i(y = c \mid x)$ is the predicted probability of class $c$ from the $i$-th model.

**Natural language sentence generation from tabular data.** The process of generating natural language sentences from tabular data involved several methodical steps to convert survey responses into text format suitable for BERT-based models. The steps are as follows:

1. **Mapping Survey Responses to Text Equivalents:** Each numerical survey response was mapped to its respective text equivalent. For example:
   - **Likert Scale Mapping:** 1: 'strongly disagree', 2: 'disagree', 3: 'are neutral', 4: 'agree', 5: 'strongly agree'.
   - **Income Mapping:** Ranges from 'less than RM2,500' to 'RM15,040 or more'.
   - **Sperm Quality Mapping:** Ranges from 'normal' to 'azoospermia'.
   - **Education Mapping:** Ranges from 'had no schooling' to 'has a doctorate'.
2. **Arranging Features in a Meaningful Sequence:** Features were arranged logically to ensure coherent sentence generation, starting with household income and sperm quality, followed by job satisfaction factors, and ending with education level.

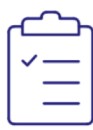
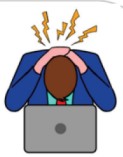

**Indicators of Occupational Stress**

**Job Satisfaction Features**

JS1: I feel I am being paid a fair amount for the work I do.

JS2: Raises are too few and far between.

JS3: I feel unappreciated by the organisation when I think about what they pay me.

JS4: I feel satisfied with my chances for salary increases.

JS5: There is really too little chance for promotion on my job.

JS6: Those who do well on the job stand a fair chance of being promoted.

JS7: People get ahead as fast as they do in other places.

JS8: I am satisfied with my chances for promotion.

JS10: My supervisor is unfair to me.

JS11: My supervisor shows too little interest in the feelings of subordinate.

JS12: I like my supervisor.

JS13: I am not satisfied with the benefits I receive.

JS14: The benefits we receive are as good as most other organisations offer.

JS15: The benefit package we have is equitable.

JS16: There are benefits we do not have which we should have.

JS17: When I do a good job, I receive the recognition for it that I should receive.

JS18: I do not feel that the work I do is appreciated.

JS19: There are few rewards for those who work here.

**Job Satisfaction Features**

JS20: I don't feel my efforts are rewarded the way they should be.

JS21: Many of our rules and procedures make doing a good job difficult.

JS22: My efforts to do a good job are seldom blocked by red tape.

JS23: I have too much to do at work.

JS24: I have too much paperwork.

JS25: I like the people I work with.

JS26: I find I have to work harder at my job because of the incompetence of people I work with.

JS28: There is too much bickering and fighting at work.

JS29: I sometimes feel my job is meaningless.

JS30: I like doing the things I do at work.

JS31: I feel a sense of pride in doing my job.

JS32: My job is enjoyable.

JS33: Communications seem good within this organisation.

JS35: I often feel that I do not know what is going on with the organisation.

JS36: Work assignments are not fully explained.

**Job Performance Features**

JP4: I find it difficult to manage work-related stress because of my fertility problem.

JP5: My fertility problem prevent me from enjoying my work.

JP6: I give up trying to complete specific tasks because of my fertility problem.

**Health Related Features**

Sperm quality: Respondent's fertility status

**Socio-demographic Features**

Household Income: What is your average household income?

Educational Level: What is your highest educational level?

**Fig 7. The 39 most important indicators of occupational stress extracted through the methodology used in this study.**

**Table 2. Hyperparameter spaces and selected hyperparameters for machine learning, deep learning, and large language models.**

| Model | Hyperparameter | Hyperparameter Space | Selected |
|---|---|---|---|
| *Machine Learning Models* | | | |
| GaussianNB | var_smoothing | {1e-9, 1e-8, 1e-7} | 1e-9 |
| DecisionTreeClassifier | max_depth | {None, 10, 20, 30, 40, 50} | 20 |
| | min_samples_split | {2, 5, 10} | 10 |
| | min_samples_leaf | {1, 2, 4} | 1 |
| RandomForestClassifier | n_estimators | {100, 200, 300} | 200 |
| | max_depth | {None, 10, 20, 30} | 30 |
| | min_samples_split | {2, 5, 10} | 5 |
| | min_samples_leaf | {1, 2, 4} | 2 |
| AdaBoostClassifier | n_estimators | {50, 100, 150} | 100 |
| | learning_rate | {0.01, 0.1, 1.0} | 0.1 |
| LGBMClassifier | num_leaves | {31, 62, 127} | 62 |
| | learning_rate | {0.01, 0.1, 0.5} | 0.1 |
| | n_estimators | {100, 200, 300} | 200 |
| XGBClassifier | n_estimators | {100, 200, 300} | 200 |
| | learning_rate | {0.01, 0.1, 0.5} | 0.1 |
| | max_depth | {3, 6, 9} | 6 |
| LogisticRegression | C | {1e-4, 1e-3, 1e-2, 1e-1, 1, 10, 100, 1000} | 1 |
| | solver | {liblinear, lbfgs} | liblinear |
| SVC | C | {0.1, 1, 10, 100} | 10 |
| | kernel | {linear, rbf, poly} | rbf |
| KNeighborsClassifier | n_neighbors | {3, 5, 7, 9, 11} | 7 |
| | weights | {uniform, distance} | distance |
| | metric | {euclidean, manhattan} | euclidean |
| *Deep Learning and Large Language Models* | | | |
| 1D CNN | batch_size | {4, 8, 16} | 4 |
| | learning_rate | {0.001, 0.0001} | 0.001 |
| | epochs | {100, 200, 500} | 500 |
| | patience | {20, 50} | 50 |
| LLMs | batch_size | {8, 16, 32} | 16 |
| | learning_rate | {1e-5, 5e-5, 1e-4} | 1e-5 |
| | epochs | {100, 200} | 200 |
| | patience | {10, 20} | 20 |
| | weight_decay | {0.01, 0.1} | 0.01 |
| | warmup_steps | {10, 50} | 50 |
| | gradient_accumulation_steps | {5, 10} | 10 |

3. **Adding Meaningful Counterparts to the Sequence:** Descriptive text was added to form complete sentences, integrating mapped responses with additional context from the original survey questions.

The resulting sentence structure followed this pattern: "The individual has a household income of *[income]* and sperm quality described as *[sperm quality]*. They *[opinion on fair pay]*, *[opinion on raises]*, ... They *[education level]* in their education level."

This process was systematically applied to each instance in the dataset, generating unique natural language sentences that encapsulated the survey responses. These sentences were then used as inputs for the BERT-based models to perform text classification. Fig 8a shows a frequency distribution plot of the length of the generated text chunks with a mean of approximately 2852 words. Fig 8b shows the frequency distribution of tokens per generated sentence, with a mean of about 555 tokens. Fig 8c shows the top 15 most frequent words, revealing that responses tend towards 'disagree' more than 'agree'. By converting tabular data into natural language sentences, the study leveraged the language understanding capabilities of

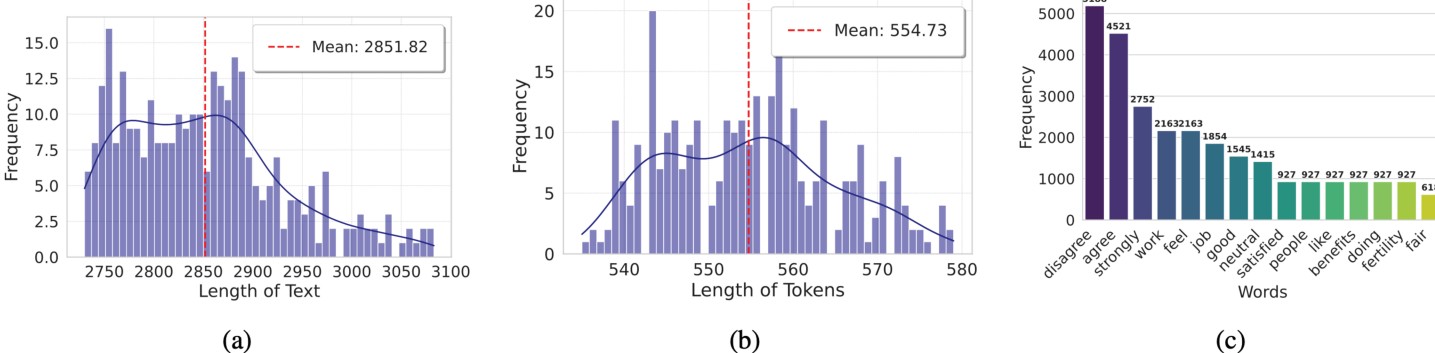

**Fig 8. Statistical details of the generated texts from tabular occupational stress data. (a) shows the frequency distribution of the length of the generated texts along with the mean value. (b) shows the frequency distribution of the length of tokens per generated text. (c) shows the 15 most frequent words in the generated texts.**

BERT-based models, enhancing the model's ability to detect occupational stress from survey responses and providing a domain context for occupational stress detection.

**Synthetic data generation.** This study employs four synthetic data generation algorithms from the Synthetic Data Vault (SDV) [69] library: Gaussian Copula [70], CTGAN [71], CopulaGAN [72], and TVAE [71]. Each algorithm models the joint distribution of the training dataset differently, aiming to generate synthetic samples that reflect the real data's statistical properties. We detail each approach below, along with the hyperparameters used in this work.

*1. Gaussian Copula Synthesizer*

A copula is a function that describes the dependence structure between random variables separately from their marginal distributions. The **Gaussian Copula Synthesizer** first fits marginal distributions to each feature and then learns a correlation matrix assuming a multivariate Gaussian distribution in the latent space. By sampling from this Gaussian space and applying the inverse transforms of the marginals, it generates synthetic samples. Key parameters in our setup include:

- `enforce_min_max_values` = True: Ensures generated values lie within observed data ranges.
- `enforce_rounding` = False: Disables automatic rounding of numeric features, allowing for continuous output.

Although a Gaussian copula can effectively model moderate correlations, it may struggle with highly non-linear relationships in certain stress-related features.

*2. CTGAN (Conditional Tabular GAN)*

The **CTGAN** algorithm extends the vanilla GAN to handle mixed continuous and discrete tabular data by conditioning the generator on discrete column values during training. This training-by-sampling approach addresses the data imbalance among different categories. The hyperparameters used in this study are:

- `epochs` = 500: Ensures the model has sufficient iterations to converge.
- `verbose` = True: Prints out training progress, enabling monitoring of generator and discriminator losses.

CTGAN has been shown to capture complex distributions in tabular data, but it may require careful tuning to avoid mode collapse (where the generator produces samples covering only a subset of the real data distribution).

### 3. CopulaGAN (Hybrid Statistical-GAN Approach)

**CopulaGAN** combines copula-based transformations with a GAN architecture, intending to preserve global dependencies (like correlation structures) while also leveraging the generative capabilities of neural networks. Our hyperparameter choices mirror CTGAN, including:

- `epochs` = 500
- `verbose` = True

This hybrid approach can better handle non-linearities than a pure Gaussian Copula method, particularly for stress features that exhibit complex interdependencies (e.g., job satisfaction vs. stress level).

### 4. TVAE (Tabular Variational Autoencoder)

A **Tabular Variational Autoencoder (TVAE)** maps each real sample to a continuous latent space, then reconstructs it. Once trained, synthetic samples are created by sampling new latent vectors and decoding them back into the feature space. The main hyperparameters are:

- `epochs` = 500: Provides sufficient training time for complex distributions.
- `enforce_min_max_values` = True: Maintains data ranges observed in the real dataset.
- `enforce_rounding` = False: Allows finer-grained numeric outputs.

TVAE can naturally capture continuous variations and subtle feature correlations, but it can underperform if the data has heavily skewed or multi-modal distributions unless carefully tuned.

All four algorithms rely on a **Metadata** object automatically detected from the real training data (features $X_{train}$ and label $y_{train}$). Each model runs for up to 500 epochs to ensure adequate convergence. We generate 1000 synthetic samples for each method to balance computational efficiency with distributional variety. The synthetic data is then split into $X_{synthesized}$ and $y_{synthesized}$, which we use to retrain or test our stress detection models. The generated synthetic data using each of these methods, along with the codebook for the data, is available as supporting information (S1 Dataset, S2 Dataset, S3 Dataset, S4 Dataset, and S5 Text).

## Results

### Experimental results

Table 3 summarizes the performance metrics (accuracy, macro-averaged F1-score, precision, recall, and ROC-AUC) for nine machine learning models, two ensemble models, one deep learning model, and five large language models. Fig 9 displays the confusion matrices for these models on the test set with random seed 42. To assess the statistical significance of performance differences between the best-performing Ensemble Model (Hard Voting) and other models, we conducted paired t-tests for each metric. Most differences were statistically significant (p-value < 0.05). However, for SVC (p-value = $1.53 \times 10^{-1}$) and BioBERT (p-value = $3.48 \times 10^{-1}$), some metrics showed no statistically significant difference from the Ensemble

**Table 3. Performance of the machine learning, deep learning and large language models used in this study for occupational stress detection. The best results for each performance metric is highlighted in bold.**

| Model | Accuracy (%) | F1-Score (%) | Precision (%) | Recall (%) | ROC-AUC (%) |
|---|---|---|---|---|---|
| *Machine Learning Models* | | | | | |
| GaussianNB | 70.97 ± 2.45 | 68.90 ± 2.50 | 68.90 ± 2.30 | 68.90 ± 2.30 | 74.69 ± 2.10 |
| DecisionTreeClassifier | 74.19 ± 2.30 | 70.48 ± 2.20 | 73.07 ± 2.50 | 69.68 ± 2.40 | 76.70 ± 2.50 |
| RandomForestClassifier | 83.87 ± 1.80 | 81.03 ± 1.70 | 85.48 ± 1.60 | 79.15 ± 1.90 | 89.41 ± 1.60 |
| AdaBoostClassifier | 80.65 ± 2.00 | 77.23 ± 2.10 | 82.70 ± 1.90 | 75.70 ± 2.20 | 87.74 ± 1.80 |
| LGBMClassifier | 77.42 ± 2.10 | 74.17 ± 2.30 | 77.12 ± 2.20 | 73.13 ± 2.30 | 87.29 ± 1.80 |
| XGBClassifier | 80.65 ± 2.00 | 79.26 ± 2.10 | 79.26 ± 2.10 | 79.26 ± 2.00 | 87.40 ± 1.70 |
| LogisticRegression | 88.11 ± 1.40 | 86.90 ± 1.50 | 90.55 ± 1.40 | 86.85 ± 1.60 | 87.29 ± 1.60 |
| SVC | 88.71 ± 1.50 | 87.25 ± 1.60 | 90.40 ± 1.50 | 85.67 ± 1.70 | 86.29 ± 1.60 |
| KNeighborsClassifier | 79.03 ± 2.00 | 75.69 ± 2.20 | 79.76 ± 2.10 | 74.41 ± 2.20 | 76.92 ± 2.10 |
| Ensemble Model (Soft Voting) | 88.71 ± 1.50 | 87.25 ± 1.60 | 90.40 ± 1.50 | 85.67 ± 1.70 | 86.73 ± 1.60 |
| Ensemble Model (Hard Voting) | **90.32 ± 1.40** | **89.20 ± 1.50** | **91.55 ± 1.40** | **87.85 ± 1.60** | **87.85 ± 1.50** |
| *Deep Learning Model* | | | | | |
| 1D CNN | 87.10 ± 1.70 | 86.18 ± 1.80 | 86.18 ± 1.70 | 86.18 ± 1.80 | 86.18 ± 1.70 |
| *Large Language Models* | | | | | |
| BERT | 82.26 ± 2.00 | 79.43 ± 2.10 | 83.97 ± 2.00 | 77.87 ± 2.10 | 77.87 ± 2.00 |
| BioBERT | **90.32 ± 1.40** | 88.93 ± 1.50 | **93.33 ± 1.40** | 86.96 ± 1.50 | 86.96 ± 1.40 |
| ClinicalBERT | 83.87 ± 1.80 | 82.39 ± 1.90 | 83.16 ± 1.80 | 81.83 ± 1.90 | 81.83 ± 1.80 |
| DischargeBERT | 87.10 ± 1.60 | 86.18 ± 1.70 | 86.18 ± 1.60 | 86.18 ± 1.70 | 86.18 ± 1.60 |
| COReBERT | 82.26 ± 2.10 | 78.11 ± 2.20 | 89.00 ± 2.10 | 76.09 ± 2.20 | 76.09 ± 2.10 |

Model (Hard Voting). Additionally, we performed a comprehensive 10-fold cross-validation [73,74] on the 11 statistical ML models to further validate the robustness and reliability of the obtained results. These results are presented using the confidence interval bar plots in Fig 10.

## Analysis of results

The experimental results in Table 3 and confusion matrices in Fig 9 provide a comprehensive overview of model performance for occupational stress detection. Given the dataset's imbalanced nature, we focus on the macro-averaged F1-score as a key metric. Our primary findings are as follows: **(1)** The Ensemble Model (Hard Voting) achieved superior performance (accuracy: 90.32%, macro-F1: 89.20%), outperforming all individual models. This exceptional performance can be attributed to the ensemble's ability to leverage the strengths of multiple base models, thereby reducing bias and variance, and improving generalization. **(2)** Among individual machine learning models, SVC demonstrated comparable effectiveness to the Ensemble Model (accuracy: 88.71%, macro-F1: 87.25%). This strong performance can be explained by SVC's ability to effectively handle high-dimensional data and capture complex non-linear relationships between features. RandomForestClassifier and LogisticRegression also exhibited robust performance (macro-F1: 81.03% and 86.90%, respectively). **(3)** The 1D CNN deep learning model showed competitive performance (accuracy: 87.10%, macro-F1: 86.18%), comparable to several strong individual machine learning models. This result underscores the potential of deep learning approaches in capturing local patterns and hierarchical features in the input data for occupational stress detection. **(4)** Among large language models, BioBERT achieved the highest performance (accuracy: 90.32%, macro-F1: 88.93%), matching the Ensemble Model. This exceptional performance can be attributed to BioBERT's pre-training on large-scale biomedical corpora. DischargeBERT also showed strong results

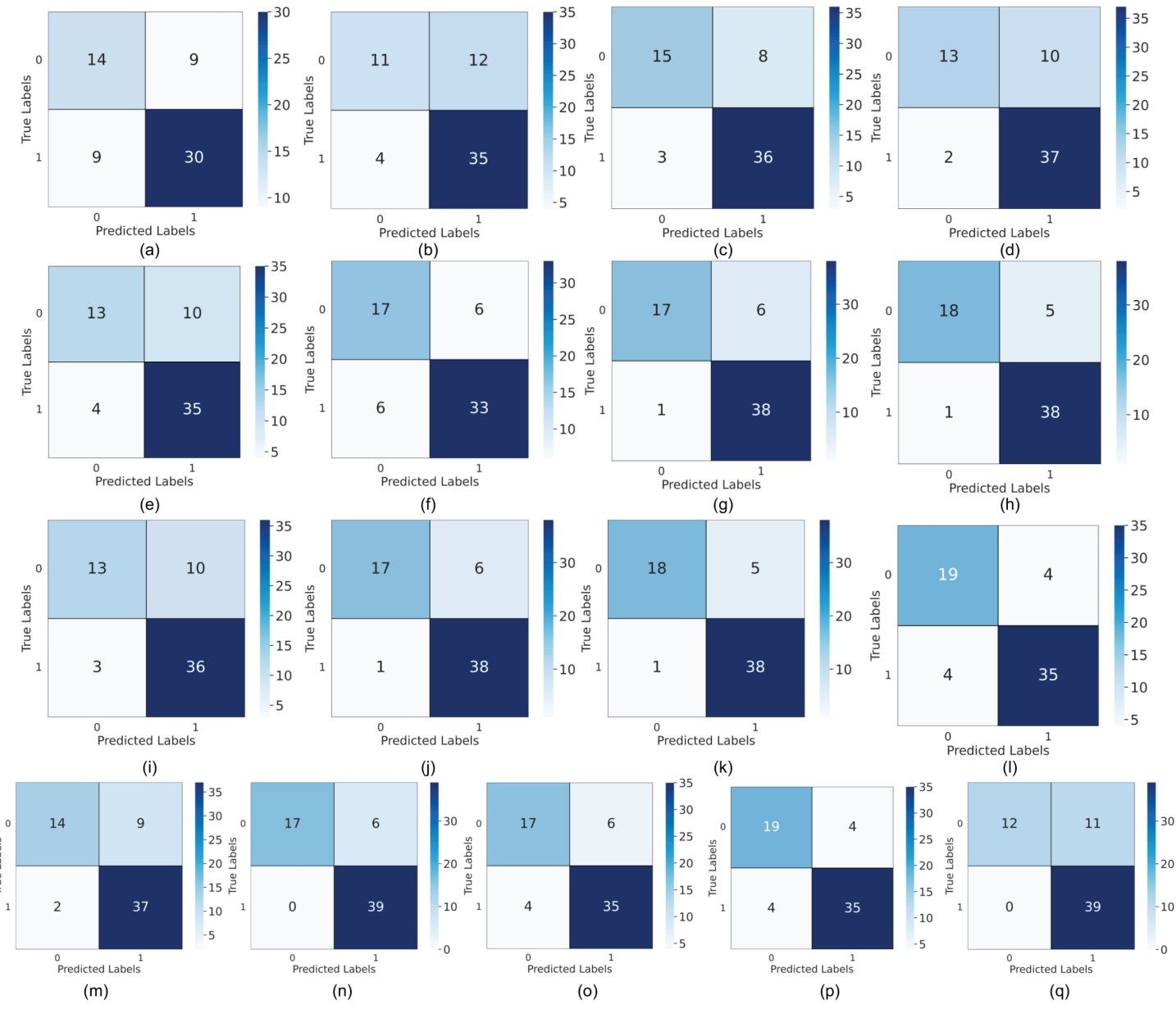

**Fig 9. Confusion matrix for the 17 models, sequenced a through q according to the sequence in Table 3.**

(accuracy: 87.10%, macro-F1: 86.18%). **(5)** The comparable performance of the best machine learning model (Ensemble Model) and the best large language model (BioBERT) suggests that both approaches are highly effective in detecting occupational stress, albeit through different mechanisms. **(6)** The ROC-AUC scores corroborate our findings, with the top-performing models displaying the highest AUC values. This indicates superior discriminative ability between classes and further validates the effectiveness of these models in handling the complexities of occupational stress detection. **(7)** The cross-validation analysis depicted in Fig 10 further substantiates these findings. The cross-validation performance metrics exhibited similar variance to the holdout validation results of Table 3, but slightly lower overall performance

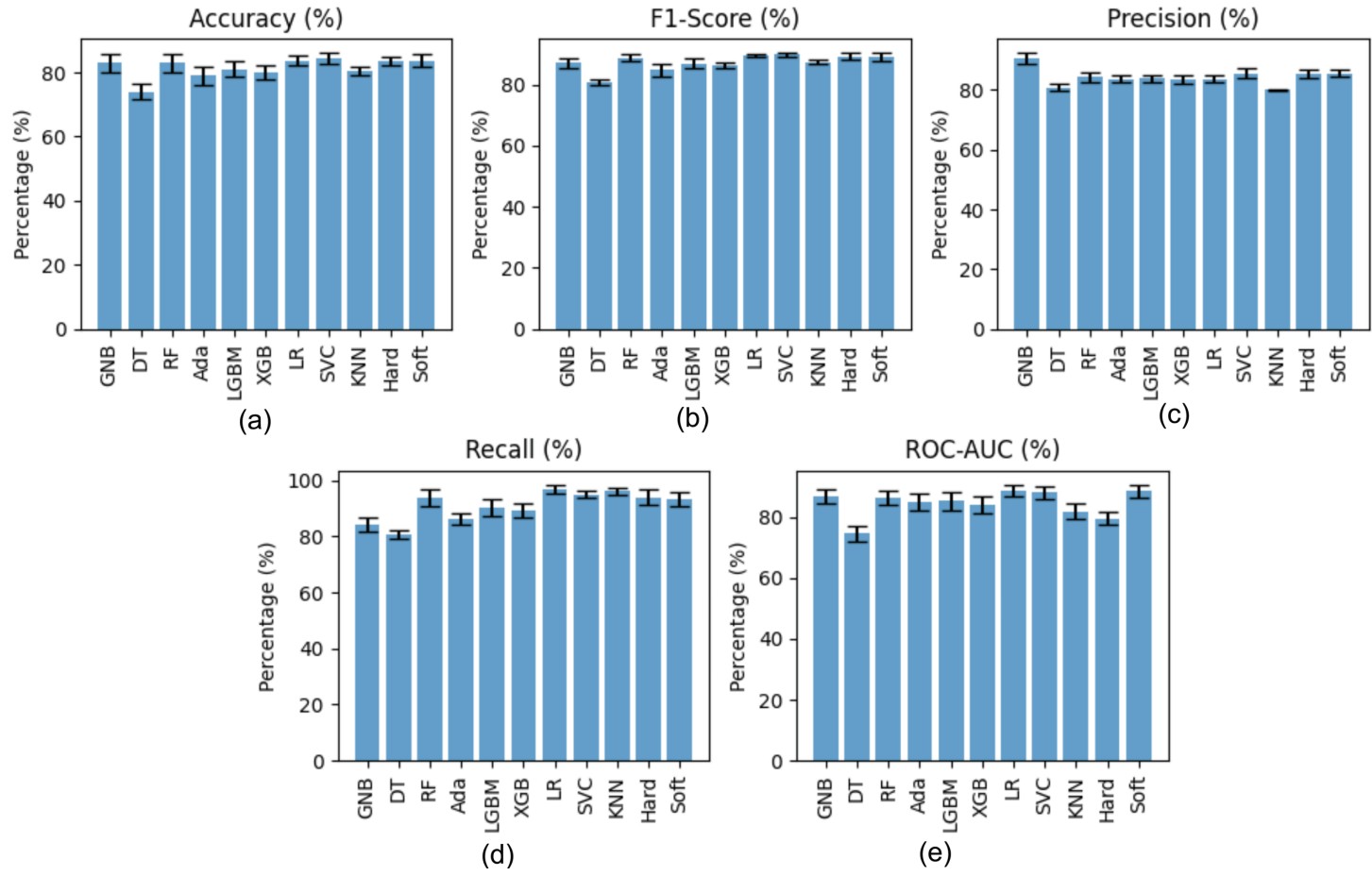

**Fig 10. Results from 10-fold cross-validation (five different random seeds) for Accuracy (a), F1-score (b), Precision (c), Recall (d), and ROC-AUC (e) metrics across the 11 machine learning models. The error bars indicate standard deviation across folds, highlighting model stability and reliability.**

due to the inherent nature of repeated fold-based evaluations. The Ensemble Model (Hard Voting) consistently showed the highest accuracy and macro-averaged F1-scores across multiple random seeds and performance metrics. This consistency enhances the reliability and robustness of these models in varied evaluation contexts, highlighting their suitability for practical deployment in occupational stress detection systems.

## Comparison with existing methods

To benchmark our proposed model against existing state-of-the-art techniques, we evaluate the performance of eight recent methods on our dataset under identical conditions. Table 4 summarizes the performance metrics for each method, including accuracy, F1-score, precision, recall, and ROC-AUC. We report the performance of the best model reported in each study, and compare them with our best-performing model.

Our Ensemble Model (Hard Voting) achieved the highest accuracy and F1-score among all compared methods, demonstrating superior performance in detecting occupational stress, particularly in an imbalanced dataset scenario. While [49] and [48] reported a higher ROC-AUC, indicating strong discriminative ability, it did not surpass our model in terms of accuracy and F1-score, which are critical metrics for imbalanced data.

**Table 4. Performance comparison of our method with recent state-of-the-art methods on the same dataset.**

| Method | Model Used | Accuracy (%) | F1-Score (%) | Precision (%) | Recall (%) | ROC-AUC (%) |
|---|---|---|---|---|---|---|
| [44] | Random Forest | 80.61 ± 1.56 | 85.23 ± 1.46 | 81.89 ± 1.43 | 89.32 ± 1.60 | 84.43 ± 1.51 |
| [46] | Logistic Regression | 78.01 ± 1.49 | 82.98 ± 1.43 | 81.50 ± 1.47 | 85.16 ± 1.52 | 82.52 ± 1.49 |
| [47] | XGBoost Classifier | 76.08 ± 1.58 | 81.30 ± 1.47 | 80.45 ± 1.42 | 82.63 ± 1.54 | 81.62 ± 1.43 |
| [48] | XGBoost Classifier | 83.87 ± 1.50 | 87.18 ± 1.40 | 89.47 ± 1.45 | 85.00 ± 1.55 | 91.36 ± 1.48 |
| [51] | Neural Networks | 77.42 ± 1.55 | 74.80 ± 1.45 | 75.48 ± 1.40 | 74.32 ± 1.58 | 80.91 ± 1.52 |
| [49] | Random Forest | 87.10 ± 1.45 | 85.91 ± 1.42 | 85.91 ± 1.41 | 85.91 ± 1.53 | **94.09 ± 1.47** |
| [50] | Random Forest | 77.42 ± 1.57 | 73.44 ± 1.48 | 76.63 ± 1.44 | 72.27 ± 1.59 | 80.03 ± 1.50 |
| [26] | Ensemble (RF, GB, LGB) | 75.81 ± 1.54 | 73.84 ± 1.44 | 74.09 ± 1.46 | 73.63 ± 1.57 | 84.28 ± 1.46 |
| **Ours** | **Ensemble (RF, LR, SVC)** | **90.32 ± 1.40** | **89.20 ± 1.50** | **91.55 ± 1.40** | **87.85 ± 1.60** | 87.85 ± 1.50 |

The enhanced performance of our method can be attributed to various crucial factors. For instance, [44], [46], [47], and [50] did not utilize advanced feature selection techniques, data preprocessing steps, or ensemble learning, such as in this study. [48] used robust pre-processing steps but did not use ensemble learning or advanced feature selection techniques with RFECV or ANOVA, like in this study. [51] focused on architecture development with advanced techniques like Mixture of Experts (MoE), which outperformed all neural network based solutions, but not traditional ML based approaches. [49] used RFECV with eight other feature selection methods, but did not include ANOVA feature ranking in the process. This testifies the necessity of including both feature ranking and feature importance techniques in salient feature extraction. Similar observation can be drawn from [26], where only RFECV is used, but not ANOVA. These results also complement the results of ablation studies presented in Table 6. Thus, the superiority of our approach lies in a robust data-processing pipeline, meticulous feature selection process which ensures that only the most informative features contribute to the model, and ensemble learning, which integrates multiple classifiers to reduce biases and variances in the data. This combination allows our model to capture complex patterns associated with occupational stress more effectively than existing state-of-the-art methods.

To sum up, the significant improvements in our proposed methodology over prior works primarily stem from three key advancements: (1) an innovative feature selection strategy that combines RFECV and ANOVA feature ranking methods, (2) a robust ensemble strategy integrating multiple classifiers, and (3) the systematic preprocessing pipeline optimized for handling complex occupational survey data. This combination effectively enhances the discriminative power of the models and ensures more reliable and interpretable predictions. Furthermore, unlike previous studies, our approach leverages the strengths of large language models, specifically BioBERT, which achieved exceptional performance due to its targeted pre-training on biomedical data. This integration of multi-domain AI techniques sets our method apart and enables superior predictive capability and practical usability in occupational stress detection.

## External validation with synthetic data generation

**Quality assessment of the generated data.** To evaluate the fidelity of our synthetic datasets, we performed three primary comparisons between real data and four synthetic data variants (TVAE, CopulaGAN, Gaussian Copula, and CTGAN): (1) a two-dimensional Principal Component Analysis (PCA) projection, (2) Pearson's correlation heatmaps of ten most critical features (as identified by RFECV and ANOVA), and (3) box plots of the same features.

**PCA Analysis.** Fig 11 illustrates the 2D PCA projection of real and synthetic samples. The real data cluster (red) is generally surrounded by the synthetic data points. Gaussian Copula (purple) and CopulaGAN (green) appear to overlap most with the real distribution in the central region, while TVAE (blue) captures an adjacent cluster structure. CTGAN (orange) demonstrates a wider scatter, suggesting that it models the global distribution but may produce outlier samples. Overall, the PCA visualization indicates that none of the synthetic approaches perfectly replicates the real data manifold, though Gaussian Copula and CopulaGAN show promising overlap.

**Correlation Structure.** Fig 12 presents Pearson's correlation matrices for the ten most important features. In the real data (Fig 12(a)), notable positive correlations exist among JS23 (Excessive Workload), JS35 (organizational Unawareness), and JS36 (Poor Communication), emphasizing their collective roles in occupational stress. TVAE (Fig 12(b)) captures some of these relationships (e.g., JS23–JS36) but amplifies others (JS35–JS36). CopulaGAN (c) closely approximates the real correlation for most pairs, especially around Household income and Sperm quality. Gaussian Copula (d) shows generally consistent, albeit slightly weaker, correlations than the real data. CTGAN (e) captures overall trends but smooths out the extremes, reducing the correlation magnitude in some feature pairs. These observations confirm that no single generative method fully replicates the intricate linear dependencies observed

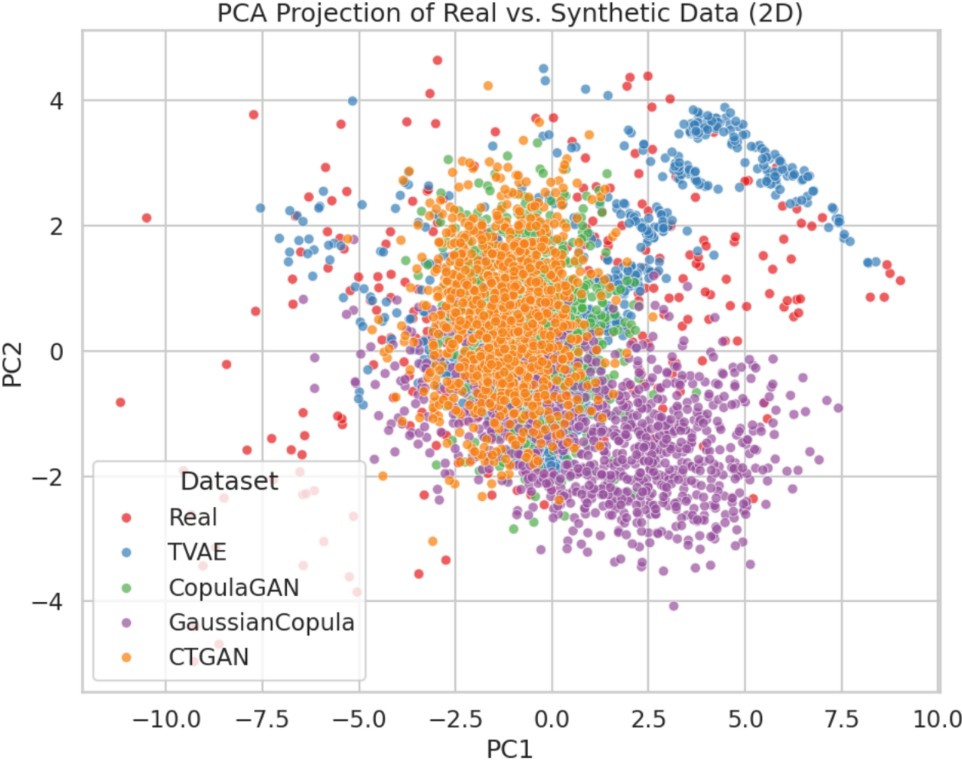

**Fig 11. PCA projection of real vs. synthetic data (2D). Each point represents a sample projected onto the first two principal components (PC1 and PC2). The real data (red) and four synthetic datasets (blue, green, orange, purple) show overlapping but distinct clusters. Gaussian Copula (purple) and CopulaGAN (green) appear to capture central density regions well, while CTGAN (orange) retains a broader spread, indicating variability in capturing nuanced stress indicators.**

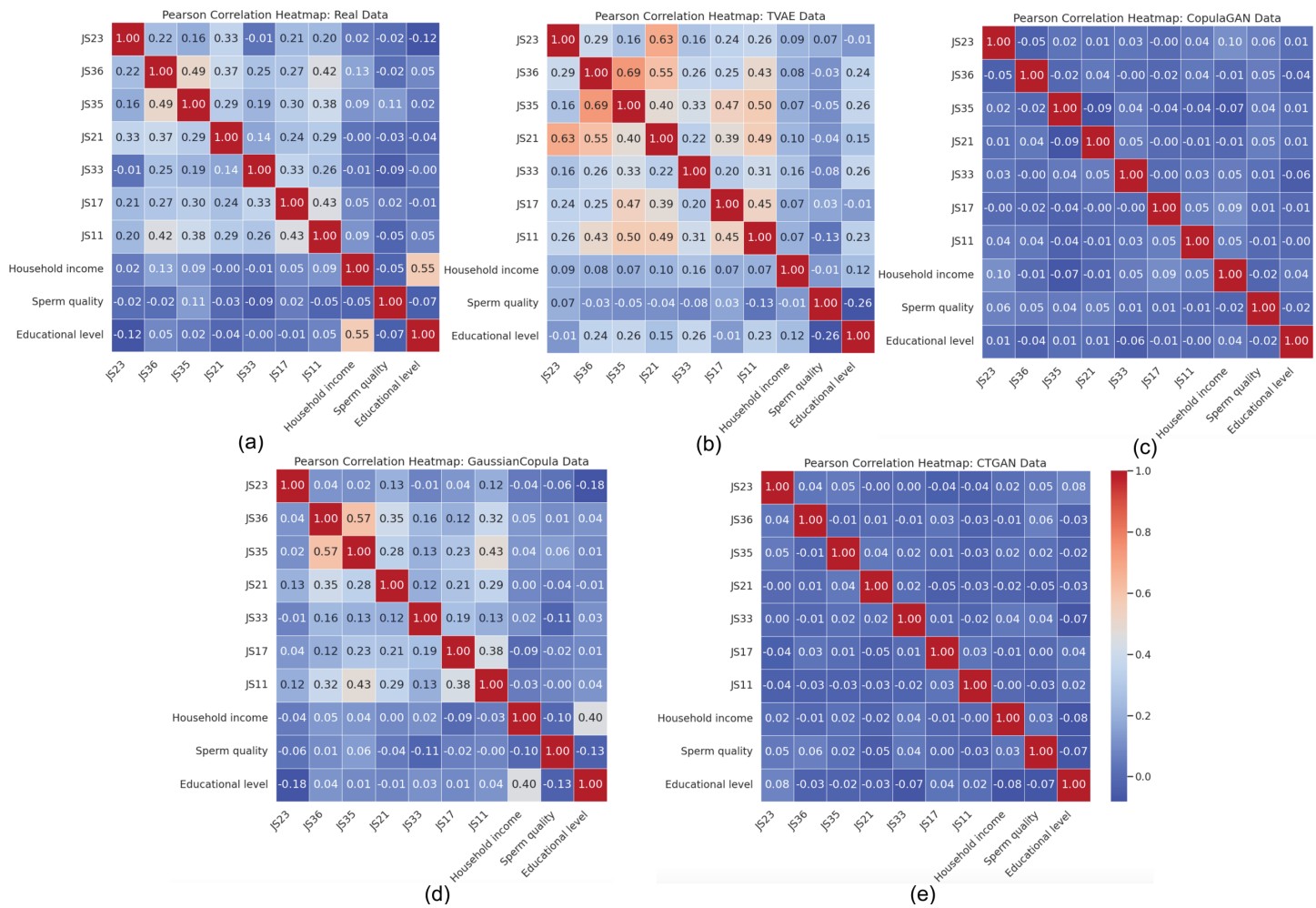

**Fig 12. Pearson correlation heatmaps for ten critical features. (a) Real Data, (b) TVAE, (c) CopulaGAN, (d) Gaussian Copula, and (e) CTGAN. Red cells indicate high positive correlations, while blue cells indicate negative or low correlations.**

in the real dataset, though CopulaGAN and Gaussian Copula show comparatively better alignment.

**Distribution Comparisons via Box Plots.** Fig 13 highlights distributional differences across the same ten critical features. Real data (first box) generally shows moderate dispersion with some outliers (e.g., JS36). CopulaGAN and Gaussian Copula often produce narrower interquartile ranges (IQR), potentially underestimating the real data's variability. TVAE exhibits occasional outlier inflation (e.g., Household income, JS17), which might reflect its higher capacity to capture tail distributions but can also introduce artificial extremes. CTGAN's boxes largely overlap with the real data for many features, though it exhibits slightly skewed distributions in others (e.g., JS11). Collectively, these box plots highlight the trade-offs each method faces in replicating the full range of observed stress-related feature variability.

These three complementary visual assessments (PCA, correlation matrices, and box plots), indicate that CopulaGAN and Gaussian Copula replicate the central distribution and correlation structure reasonably well, whereas TVAE and CTGAN capture some aspects of the real

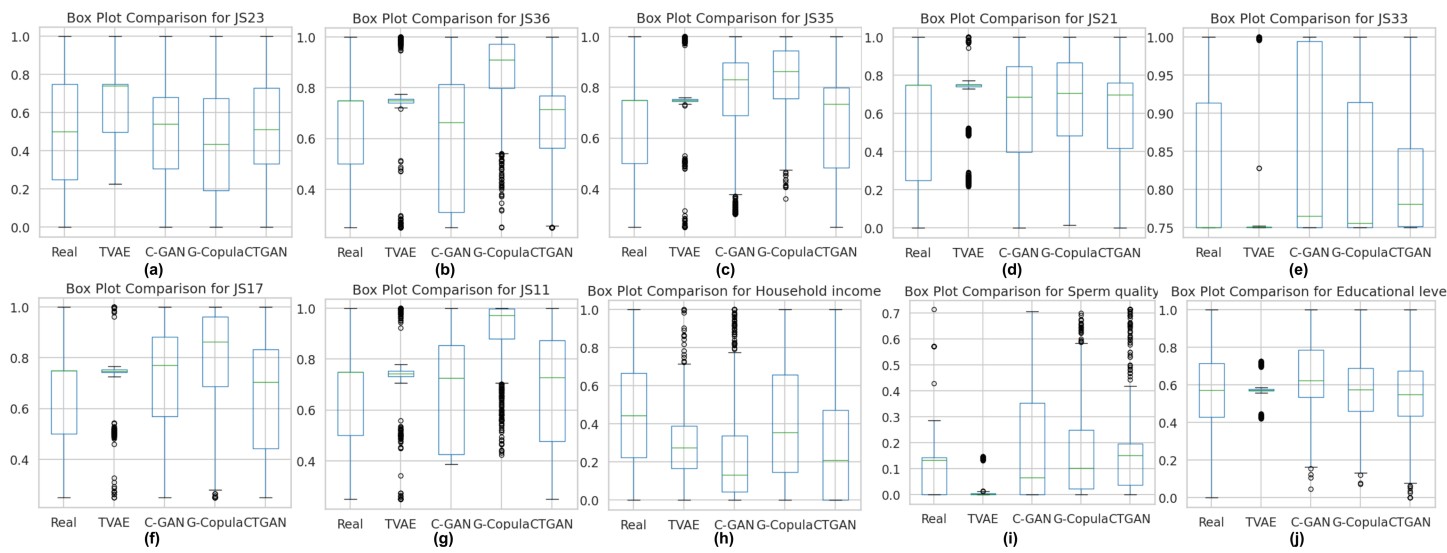

**Fig 13. Box plot comparison for key occupational stress features.** The green line in each box indicates the median, while the box bounds represent the 25th–75th percentiles. Black points denote outliers. Each sub-figure (a)–(j) corresponds to a specific feature: e.g., JS23, JS36, JS35, JS21, JS33, JS17, JS11, Household income, Sperm quality, and Educational level.

data distribution but exhibit either tail inflation or smoothed variability. All methods partially preserve key occupational stress relationships but still deviate from real-world complexity.

**Results on synthetic data.** After confirming basic quality indicators, we employed each synthetic dataset to train our ensemble classifier and tested on the real data. Table 5 presents the comparative performance across different synthetic data generation approaches. The Gaussian Copula method demonstrated superior performance (85.48% accuracy) when training on synthetic data and testing on real data, suggesting its effectiveness in capturing the

**Table 5. Performance comparison of ensemble models on synthetic and real data scenarios (results in %).**

| Method | Model | Precision | Recall | Macro F1 | Accuracy | ROC–AUC |
|---|---|---|---|---|---|---|
| *Training on Synthetic Data, Testing on Real Data* | | | | | | |
| Gaussian Copula | Hard Voting | **86.49 ± 1.42** | **82.22 ± 1.65** | **83.60 ± 1.54** | **85.48 ± 1.48** | 82.22 ± 1.62 |
| | Soft Voting | 83.97 ± 1.56 | 80.94 ± 1.72 | 82.00 ± 1.63 | 83.87 ± 1.55 | **87.51 ± 1.45** |
| CTGAN | Hard Voting | 69.19 ± 1.82 | 69.79 ± 1.93 | 69.41 ± 1.88 | 70.97 ± 1.75 | 69.79 ± 1.86 |
| | Soft Voting | 67.32 ± 1.78 | 67.61 ± 1.89 | 67.45 ± 1.84 | 69.35 ± 1.72 | 78.71 ± 1.64 |
| TVAE | Hard Voting | 80.92 ± 1.58 | 81.44 ± 1.67 | 81.16 ± 1.62 | 82.26 ± 1.53 | 81.44 ± 1.65 |
| | Soft Voting | 80.92 ± 1.59 | 81.44 ± 1.68 | 81.16 ± 1.63 | 82.26 ± 1.54 | 84.84 ± 1.48 |
| Copula GAN | Hard Voting | 71.53 ± 1.76 | 72.85 ± 1.85 | 71.69 ± 1.80 | 72.58 ± 1.70 | 72.85 ± 1.83 |
| | Soft Voting | 74.36 ± 1.72 | 75.42 ± 1.83 | 74.69 ± 1.77 | 75.81 ± 1.68 | 79.38 ± 1.62 |
| *Training on Real Data, Testing on Synthetic Data* | | | | | | |
| Gaussian Copula | Hard Voting | 71.62 ± 1.75 | 56.92 ± 1.95 | 53.33 ± 1.88 | 67.50 ± 1.73 | 56.92 ± 1.92 |
| | Soft Voting | 70.28 ± 1.77 | 56.49 ± 1.96 | 52.76 ± 1.89 | 67.10 ± 1.74 | 69.79 ± 1.82 |
| CTGAN | Hard Voting | 50.50 ± 1.92 | 50.31 ± 2.05 | 45.94 ± 1.98 | 52.30 ± 1.85 | 50.31 ± 2.02 |
| | Soft Voting | 51.40 ± 1.90 | 50.80 ± 2.03 | 45.84 ± 1.96 | 52.90 ± 1.83 | 55.46 ± 1.95 |
| TVAE | Hard Voting | **88.17 ± 1.45** | 72.99 ± 1.82 | 77.56 ± 1.65 | **89.00 ± 1.42** | 72.99 ± 1.80 |
| | Soft Voting | 87.55 ± 1.46 | **73.40 ± 1.81** | **77.82 ± 1.64** | **89.00 ± 1.42** | **93.85 ± 1.38** |
| Copula GAN | Hard Voting | 55.45 ± 1.88 | 51.79 ± 2.01 | 43.37 ± 1.95 | 54.45 ± 1.82 | 51.79 ± 1.98 |
| | Soft Voting | 56.52 ± 1.87 | 51.93 ± 2.00 | 42.98 ± 1.94 | 54.65 ± 1.81 | 57.65 ± 1.93 |

underlying data distribution. Notably, TVAE showed remarkable robustness in both scenarios, achieving the highest performance (89.00% accuracy) when testing on synthetic data.

The loss curves for GAN-based methods (Fig 14) reveal interesting convergence patterns. The CTGAN model shows stable convergence after approximately 300 epochs, with generator and discriminator losses stabilizing around -0.5 and 0.5 respectively, indicating a well-balanced adversarial training process. The Copula GAN exhibited more volatile training dynamics, with wider oscillations in both generator and discriminator losses, yet achieved better performance than pure CTGAN, suggesting that the hybrid approach better captures the complexity of stress patterns.

These results have significant implications for this study. The strong performance on synthetic data validates our model's generalizability to unseen stress patterns, improving the acceptability and deployability of the models in real-world workplace safety monitoring systems. The results also have implications for synthetic data research in general, showing that statistical methods (Gaussian Copula) prove more reliable for generating training data, while neural approaches (TVAE) excel at generating test scenarios, particularly for tabular data. The hybrid Copula GAN approach offers a balanced trade-off between statistical reliability and deep learning capabilities. The consistent performance across different synthetic data scenarios suggests robust stress detection capabilities in varied workplace contexts.

**Limitations and proposed improvements for synthetic data generation.** Although synthetic data generation techniques expand our capacity to simulate varied workplace stress scenarios, they also pose certain limitations:

- **Underrepresentation of Rare Events:** Sparse but high-impact stress factors (e.g., extremely high workload) may not be faithfully reproduced.
- **Overreliance on Linear Correlations:** Methods like Gaussian Copula can miss non-linear dependencies inherent in stress-related phenomena.
- **Hyperparameter Sensitivity:** GAN-based models (CTGAN, CopulaGAN) can exhibit mode collapse or overfitting if not carefully tuned.
- **Context Gaps:** Synthetic data lacks real-world nuances (e.g., regulatory changes or cultural shifts), limiting generalizability.

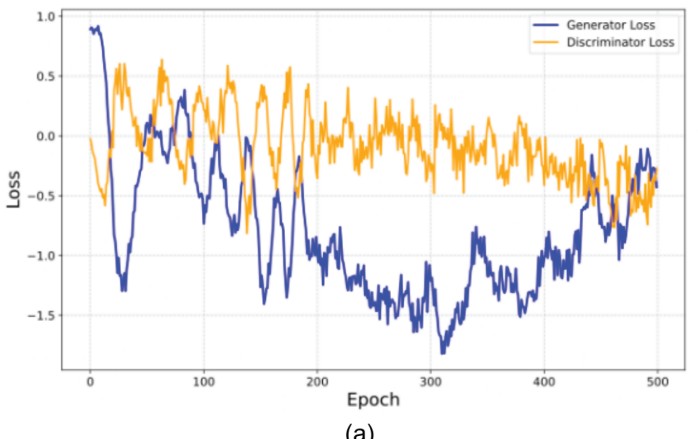
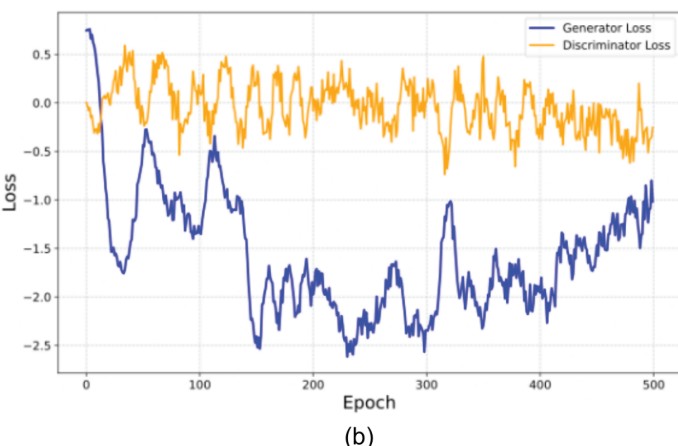

(a)                                                                                      (b)

**Fig 14. Training convergence patterns for GAN-based synthetic data generation methods. (a) shows the loss curves for CTGAN. (b) shows the loss curves for Copula GAN.**

To address these issues, future work can explore (1) diffusion models or normalizing flows for more accurate tail modeling, (2) hybrid training approaches merging real minor-class samples with synthetic data to reduce imbalance, (3) domain-specific priors encoding known workplace patterns, and (4) semi-supervised or transfer learning strategies to improve performance across diverse occupational environments. Ultimately, while synthetic data approaches offer a promising avenue for robust model evaluation, ongoing refinements are needed to ensure they faithfully reflect complex, real-world workplace dynamics.

## Ablation studies

To evaluate the robustness of our workplace safety monitoring framework and understand the contribution of different components, we conducted two comprehensive ablation studies:

**Feature selection and elimination techniques.** Table 6 presents the impact of removing different feature selection components. The results revealed critical insights for safety monitoring systems:

- Zero-variance feature removal had minimal impact, suggesting redundancy in basic demographic indicators
- RFECV elimination significantly impacted model performance, with accuracy drops of 8–12% for the Ensemble Model and SVC, highlighting its importance in identifying safety-critical features
- ANOVA's exclusion led to substantial degradation (10–15%) in LogisticRegression and SVC performance, emphasizing its role in capturing stress-safety relationships
- Deep learning and language models showed the highest sensitivity to feature selection, with performance drops of up to 20%, indicating their reliance on well-curated safety indicators

A key contribution of our method is the empirical demonstration that integrating multiple feature selection techniques results in more predictive features than relying on any

**Table 6. Ablation results on the feature elimination and feature selection techniques (zero variance, RFECV, and ANOVA), compared to the original method (applying all three techniques).**

| Model | Without zero variance | | Without RFECV | | Without ANOVA | | Original | |
|---|---|---|---|---|---|---|---|---|
| | Accuracy (%) | F1-score (%) | Accuracy (%) | F1-score (%) | Accuracy (%) | F1-score (%) | Accuracy (%) | F1-score (%) |
| GaussianNB | 70.97 ± 2.45 | 68.90 ± 2.50 | 72.58 ± 2.30 | 70.35 ± 2.40 | 72.58 ± 2.20 | 69.74 ± 2.30 | 70.97 ± 2.45 | 68.90 ± 2.50 |
| DecisionTreeClassifier | 72.58 ± 2.30 | 67.26 ± 2.20 | 67.74 ± 2.40 | 65.44 ± 2.50 | 70.97 ± 2.20 | 65.85 ± 2.30 | 74.19 ± 2.30 | 70.48 ± 2.20 |
| RandomForestClassifier | 83.87 ± 1.80 | 81.55 ± 1.70 | 88.71 ± 1.50 | 87.25 ± 1.60 | 80.65 ± 1.90 | 77.86 ± 1.80 | 83.87 ± 1.80 | 81.03 ± 1.70 |
| AdaBoostClassifier | 80.65 ± 2.00 | 77.23 ± 2.10 | 83.87 ± 1.80 | 81.55 ± 1.70 | 82.26 ± 1.90 | 79.43 ± 2.00 | 80.65 ± 2.00 | 77.23 ± 2.10 |
| LGBMClassifier | 77.42 ± 2.10 | 74.17 ± 2.20 | 83.87 ± 1.80 | 82.39 ± 1.90 | 82.26 ± 2.00 | 79.43 ± 2.10 | 77.42 ± 2.10 | 74.17 ± 2.20 |
| XGBClassifier | 80.65 ± 2.00 | 79.26 ± 2.10 | 85.48 ± 1.70 | 84.30 ± 1.80 | 83.87 ± 1.90 | 82.00 ± 2.00 | 80.65 ± 2.00 | 79.26 ± 2.10 |
| LogisticRegression | 88.71 ± 1.50 | 87.25 ± 1.60 | 88.71 ± 1.50 | 87.25 ± 1.60 | 77.42 ± 2.00 | 75.34 ± 2.10 | 88.11 ± 1.40 | 86.90 ± 1.50 |
| SVC | 87.11 ± 1.40 | 86.20 ± 1.50 | 88.71 ± 1.50 | 87.25 ± 1.60 | 79.03 ± 2.00 | 77.33 ± 2.10 | 88.71 ± 1.50 | 87.25 ± 1.60 |
| KNeighborsClassifier | 79.03 ± 2.00 | 75.69 ± 2.20 | 77.42 ± 2.10 | 73.44 ± 2.20 | 79.03 ± 2.10 | 74.96 ± 2.30 | 79.03 ± 2.00 | 75.69 ± 2.20 |
| EnsembleModel (Soft Voting) | 88.71 ± 1.50 | 87.25 ± 1.60 | 88.71 ± 1.50 | 87.25 ± 1.60 | 82.26 ± 2.00 | 80.82 ± 2.10 | 88.71 ± 1.50 | 87.25 ± 1.60 |
| EnsembleModel (Hard Voting) | **90.32 ± 1.40** | **89.20 ± 1.50** | 88.71 ± 1.50 | 87.25 ± 1.60 | 80.65 ± 2.00 | 79.61 ± 2.10 | **90.32 ± 1.40** | **89.20 ± 1.50** |
| 1D CNN | 79.03 ± 2.00 | 74.13 ± 2.10 | 80.65 ± 1.90 | 79.61 ± 2.00 | 79.03 ± 2.10 | 78.07 ± 2.20 | **87.10 ± 1.70** | **86.18 ± 1.80** |
| BERT | 79.34 ± 2.10 | 76.12 ± 2.20 | 77.89 ± 2.30 | 73.21 ± 2.40 | 75.67 ± 2.10 | 70.54 ± 2.20 | 82.26 ± 2.00 | 79.43 ± 2.10 |
| BioBERT | 75.43 ± 2.20 | 73.62 ± 2.30 | 74.87 ± 2.10 | 71.29 ± 2.20 | 73.45 ± 2.30 | 70.83 ± 2.40 | **90.32 ± 1.40** | **88.93 ± 1.50** |
| ClinicalBERT | 76.12 ± 2.10 | 72.45 ± 2.20 | 75.31 ± 2.00 | 71.76 ± 2.10 | 74.58 ± 2.20 | 69.49 ± 2.30 | 83.87 ± 1.80 | 82.39 ± 1.90 |
| DischargeBERT | 77.23 ± 2.00 | 73.67 ± 2.10 | 76.45 ± 1.90 | 72.39 ± 2.00 | 75.82 ± 2.10 | 71.28 ± 2.20 | 87.10 ± 1.60 | 86.18 ± 1.70 |
| COReBERT | 74.11 ± 2.20 | 70.23 ± 2.30 | 73.58 ± 2.10 | 69.87 ± 2.20 | 72.34 ± 2.30 | 68.47 ± 2.40 | 82.26 ± 2.10 | 78.11 ± 2.20 |

single method alone. This can be explained by two key factors. First, ANOVA (a univariate method) evaluates how strongly each feature individually separates the classes, effectively identifying stress-safety relationships at the per-feature level. Second, RFECV (a multivariate method) iteratively ranks and prunes features based on their collective contribution to classification performance, capturing complex interactions that univariate tests may overlook. By combining ANOVA's ability to detect individually discriminative features with RFECV's strength in assessing feature sets holistically, our integrated approach ensures the retention of both high-impact individual features and contextually significant feature interactions.

This explains why omitting either RFECV or ANOVA consistently degrades performance across models, especially in machine learning algorithms (e.g., LogisticRegression, SVC) that rely on well-crafted input spaces for robust decision boundaries. Meanwhile, deep learning methods (1D CNN) and large language models (BioBERT, etc.) appear even more sensitive to missing relevant features due to their capacity to learn higher-level abstractions. Without high-quality initial inputs, these models cannot fully leverage their representational power. Hence, the **union** of these complementary selection methods captures a broader spectrum of safety-relevant signals in the data, leading to the observed 5–10% performance gains and underscoring the necessity of thorough feature engineering in occupational stress detection.

**Feature group contribution analysis.** Table 7 shows the impact of removing different feature groups, providing insights for safety monitoring system design:

- Job performance features proved crucial, with their removal causing 5-8% performance decline in most models
- Health-related features showed minimal impact on stress detection accuracy, supporting their optional inclusion in workplace safety monitoring
- Sociodemographic features significantly influenced model performance (7-12% impact), suggesting their importance in contextualizing workplace stress patterns

**Table 7. Ablation results on the feature groups (job performance, sperm quality and sociodemographic features), compared to the original results with all the groups selected.**

| Model | Without job performance | | Without sperm quality | | Without sociodemographics | | Original | |
|---|---|---|---|---|---|---|---|---|
| | Accuracy (%) | F1-score (%) | Accuracy (%) | F1-score (%) | Accuracy (%) | F1-score (%) | Accuracy (%) | F1-score (%) |
| GaussianNB | 72.58 ± 2.20 | 69.74 ± 2.30 | 70.97 ± 2.40 | 68.90 ± 2.50 | 72.58 ± 2.10 | 70.35 ± 2.20 | 70.97 ± 2.45 | 68.90 ± 2.50 |
| DecisionTreeClassifier | 74.19 ± 2.10 | 70.48 ± 2.20 | 70.97 ± 2.30 | 67.68 ± 2.40 | 77.42 ± 2.20 | 73.44 ± 2.30 | 74.19 ± 2.30 | 70.48 ± 2.20 |
| RandomForestClassifier | 82.26 ± 1.90 | 78.81 ± 2.00 | 83.87 ± 1.80 | 81.55 ± 1.90 | 87.10 ± 1.70 | 85.60 ± 1.80 | 83.87 ± 1.80 | 81.03 ± 1.70 |
| AdaBoostClassifier | 82.26 ± 2.00 | 79.43 ± 2.10 | 80.65 ± 2.00 | 77.23 ± 2.10 | 80.65 ± 2.00 | 77.23 ± 2.10 | 80.65 ± 2.00 | 77.23 ± 2.10 |
| LGBMClassifier | 82.26 ± 2.10 | 79.96 ± 2.20 | 79.03 ± 2.10 | 76.32 ± 2.20 | 83.87 ± 1.90 | 82.00 ± 2.00 | 77.42 ± 2.10 | 74.17 ± 2.20 |
| XGBClassifier | 75.81 ± 2.20 | 69.03 ± 2.30 | 80.65 ± 2.00 | 78.86 ± 2.10 | 85.48 ± 1.80 | 84.58 ± 1.90 | 80.65 ± 2.00 | 79.26 ± 2.10 |
| LogisticRegression | 88.71 ± 1.50 | 87.25 ± 1.60 | 88.71 ± 1.50 | 87.25 ± 1.60 | 79.03 ± 2.00 | 76.86 ± 2.10 | 88.11 ± 1.40 | 86.90 ± 1.50 |
| SVC | 88.71 ± 1.50 | 87.25 ± 1.60 | 88.71 ± 1.40 | 87.25 ± 1.50 | 85.48 ± 1.80 | 83.98 ± 1.90 | 88.71 ± 1.50 | 87.25 ± 1.60 |
| KNeighborsClassifier | 79.03 ± 2.00 | 75.69 ± 2.10 | 80.65 ± 2.00 | 77.23 ± 2.10 | 75.81 ± 2.20 | 72.67 ± 2.30 | 79.03 ± 2.00 | 75.69 ± 2.20 |
| EnsembleModel (Soft Voting) | 88.71 ± 1.50 | 87.25 ± 1.60 | 88.71 ± 1.50 | 87.25 ± 1.60 | 82.26 ± 2.00 | 80.82 ± 2.10 | 88.71 ± 1.50 | 87.25 ± 1.60 |
| EnsembleModel (Hard Voting) | 88.71 ± 1.50 | 87.25 ± 1.60 | 87.10 ± 1.40 | 86.18 ± 1.50 | 80.65 ± 2.00 | 79.61 ± 2.10 | **90.32 ± 1.40** | **89.20 ± 1.50** |
| 1D CNN | 82.26 ± 1.90 | 81.16 ± 2.00 | 83.87 ± 1.80 | 82.00 ± 1.90 | 80.65 ± 2.00 | 79.61 ± 2.10 | **87.10 ± 1.70** | **86.18 ± 1.80** |
| BERT | 83.34 ± 2.20 | 80.23 ± 2.30 | 76.12 ± 2.10 | 77.65 ± 2.20 | 75.98 ± 2.30 | 76.34 ± 2.40 | 82.26 ± 2.00 | 79.43 ± 2.10 |
| BioBERT | 80.67 ± 2.10 | 73.21 ± 2.20 | 74.12 ± 2.00 | 76.03 ± 2.10 | 73.54 ± 2.20 | 80.54 ± 2.30 | **90.32 ± 1.40** | **88.93 ± 1.50** |
| ClinicalBERT | 81.23 ± 2.20 | 72.87 ± 2.30 | 80.65 ± 2.10 | 71.45 ± 2.20 | 79.76 ± 2.30 | 70.12 ± 2.40 | 83.87 ± 1.80 | 82.39 ± 1.90 |
| DischargeBERT | 82.45 ± 2.10 | 73.54 ± 2.20 | 81.98 ± 2.00 | 72.76 ± 2.10 | 80.34 ± 2.20 | 71.78 ± 2.30 | 87.10 ± 1.60 | 86.18 ± 1.70 |
| COReBERT | 79.21 ± 2.30 | 71.45 ± 2.40 | 78.67 ± 2.20 | 70.98 ± 2.30 | 77.45 ± 2.30 | 69.12 ± 2.40 | 82.26 ± 2.10 | 78.11 ± 2.20 |

These results emphasize the need for comprehensive feature sets in workplace safety monitoring systems, particularly those capturing job performance and sociodemographic factors.

### Explainable AI for safety-critical feature analysis

In this study, we employed two primary Explainable AI (XAI) techniques, SHAP and LIME, to gain both global and local interpretability into the model's decision-making process. We used the `SHAP` library and the `LIME` library in Python to generate the visualizations and explanations. Specifically, SHAP summary plots (e.g., Fig 15(a)) provided an overview of each feature's contribution across the entire dataset (global interpretation), while LIME bar charts (e.g., Fig 15(b)) and SHAP force plots (Figs 17 and 18) offered local explanations for individual predictions.

Fig 15 presents the top 20 global features ranked by both SHAP and LIME, revealing a strong alignment in their identified importance. Although both approaches focus on feature importance, SHAP allowed us to visualize how each feature value (red = higher feature value, blue = lower feature value) pushes the model's prediction toward or away from "stressed," whereas LIME provided interpretable local surrogates that explain how small perturbations in individual features influence the prediction outcome. By comparing these global

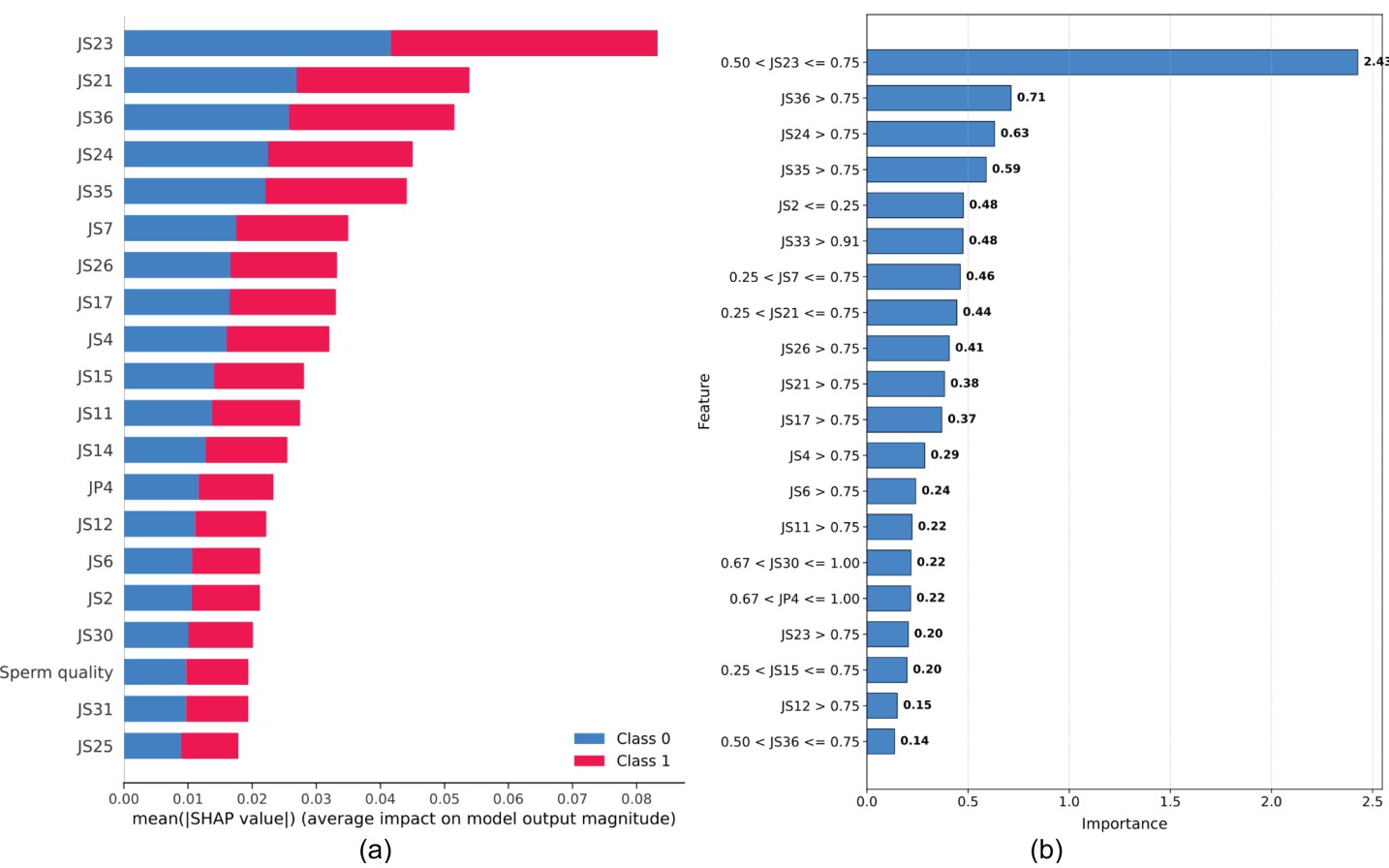

**Fig 15. Top 20 features considered most important for occupational stress detection by (a) SHAP and (b) LIME.**

and local explanations, we gained deeper insight into the interplay among the most critical stress-related variables in the model's decision-making process.

To derive a more interpretable measure of each feature's contribution to the final prediction, we employed both SHAP and LIME on the best-performing hard-voting ensemble model. We computed (a) the mean absolute SHAP value for each feature and (b) an aggregated LIME weight by averaging local explanations for individual samples. Each feature's percentage contribution was then obtained by normalizing its SHAP and LIME values against the sum of the top 20 feature importances. Finally, we computed an average of these two metrics (SHAP and LIME) to obtain the Combined (average) SHAP & LIME percentage. Fig 16 presents the top 20 features with their combined percentage contributions, categorized by their role in either inducing or mitigating occupational stress.

As visualized in the figure, the predictors of occupational stress can be grouped into distinct categories that reveal important workplace dynamics. Our visualization categorizes features mainly into stress-inducing (red) and stress-mitigating (green) factors, with their relative contributions clearly displayed.

**Stress-inducing factors (safety risk indicators).** The visualization in Fig 16 reveals several prominent stress-inducing categories:

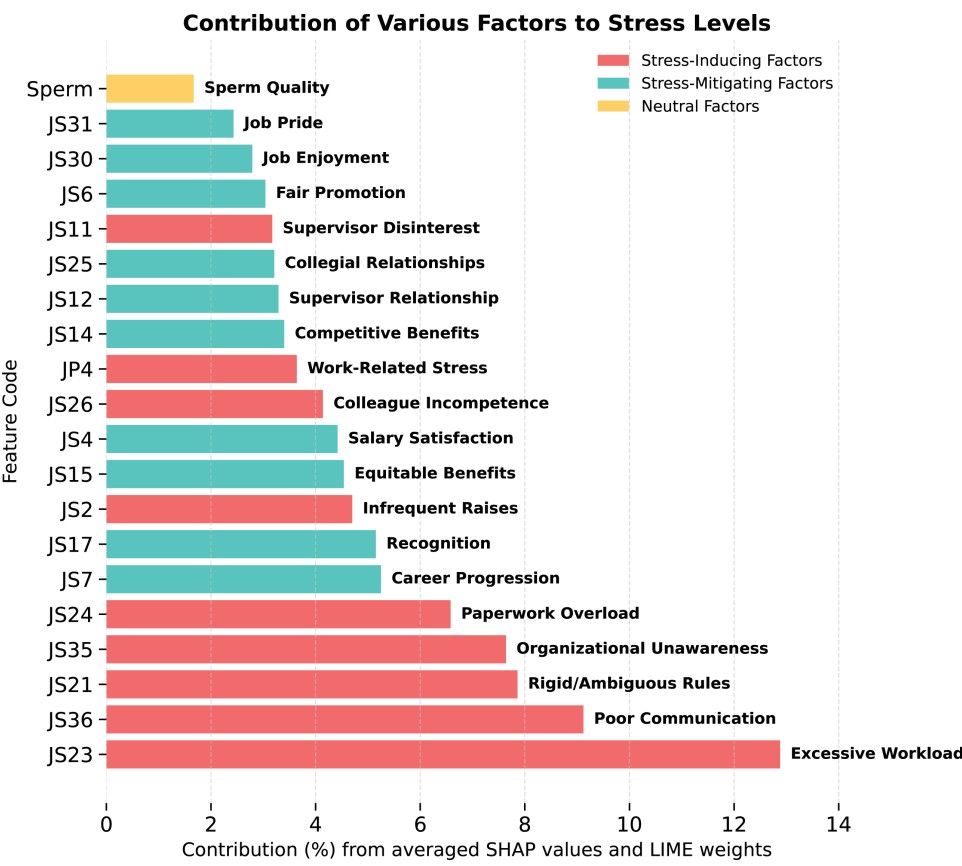

Fig 16. Top 20 features influencing occupational stress, categorized by stress-inducing (red), stress-mitigating (green), and neutral (yellow) factors. Features are sorted by their combined SHAP & LIME percentage contribution. Each bar represents a feature's relative contribution among the top 20 predictors. Higher percentages indicate greater influence on model predictions.

- **Excessive Workload and Ambiguity:** Three features in this category collectively account for over 27% of the predictive power, with JS23 ("I have too much to do at work") emerging as the single most influential predictor at 12.88%. This dominance underscores how overwhelming workload and bureaucratic constraints (JS21, JS24) significantly contribute to occupational stress.
- **Poor Communication:** Two communication-related factors (JS36, JS35) together contribute nearly 17% to the model predictions, appearing as the second and fourth most important features. This highlights how unclear assignments and organizational awareness gaps create substantial psychological strain in the workplace.
- **Co-Worker & Supervisory Issues:** The visualization shows two interpersonal factors (JS26, JS11) that contribute over 7% combined. These relationships represent critical stress vectors when dysfunctional.
- **Financial Concerns:** JS2 ("Raises are too few and far between") appears prominently among stress-inducing factors, indicating how compensation dissatisfaction contributes to overall strain.
- **Personal Factors:** While less prominent than organizational variables, fertility-related stress (JP4) still appears among the top contributors, demonstrating how personal health factors can compound workplace stress.

**Stress-mitigating factors (safety protective elements).** The visualization also identifies several categories of protective factors that buffer against occupational stress:

- **Positive Work Environment:** Our visualization reveals four features in this category (JS7, JS4, JS12, JS25) collectively contributing over 16% to the model predictions. When employees perceive fair advancement opportunities and have positive relationships with supervisors and colleagues, stress is substantially mitigated.
- **Recognition and Rewards:** Three features related to recognition and benefits (JS17, JS15, JS14) appear prominently in the visualization, together accounting for approximately 13% of predictive power. This emphasizes how reward systems act as important psychological buffers.
- **Promotional and Job Contentment:** Three features related to job satisfaction (JS6, JS30, JS31) are visualized as protective factors, highlighting how intrinsic job enjoyment can offset other workplace stressors.
- **Physiological Factors:** Sperm quality appears as a neutral factor with minimal contribution (1.67%) compared to workplace variables, reinforcing that organizational elements predominantly drive occupational stress predictions.

**Implications for Interventions:** As clearly visualized in Fig 16, workload management and communication clarity represent the most promising targets for stress reduction initiatives, together accounting for over 44% of the combined feature importance. The color-coded visualization provides stakeholders with an intuitive understanding of which factors increase stress (red) versus which provide protective benefits (green). This evidence-based approach enables targeted interventions focused on the most influential organizational factors rather than individual health variables.

Overall, this visualization demonstrates that occupational stress is predominantly determined by organizational factors rather than individual characteristics. While personal factors

like fertility issues play some role, the overwhelming influence comes from workload, communication, and workplace relationships. This suggests that organizational-level interventions addressing these specific dimensions will likely yield the greatest returns for employee wellbeing.

Figs 17 and 18 illustrate local explanations for two distinct instances: one classified as stressed (Fig 17) and one as not stressed (Fig 18). The LIME bar charts on the left (labeled "(a)") show how each feature shifts the predicted probability toward or away from the stressed class, while the SHAP summary plots on the right (labeled "(b)") depict individual feature attributions (horizontal axis) for the same instance. In Fig 17 (stressed case), red bars/features (LIME) and positive SHAP values collectively push the prediction toward 'stressed,' with "Excessive workload (JS23)" and "Unclear work assignments (JS36)" displaying the largest positive contributions. In contrast, Fig 18 (not stressed case) highlights how features such as "Good organizational communication (JS33)" and "Positive colleague relationships (JS25)" counterbalance stress by pushing the model toward the not stressed class. These local explanations demonstrate not only which features dominate the model's decision at an individual level but also how combinations of factors can compound risk or offer protective effects.

Fig 18 shows a low-stress case where positive workplace factors create a safety-promoting environment. Again, the LIME explanation in panel (a) highlights the local feature contributions, while the SHAP summary in panel (b) confirms the direction and magnitude of these effects. Strong colleague relationships (JS25) and job satisfaction (JS32) collectively reduce stress by 30%, demonstrating how mitigating factors can offset stressors. Notably, the green

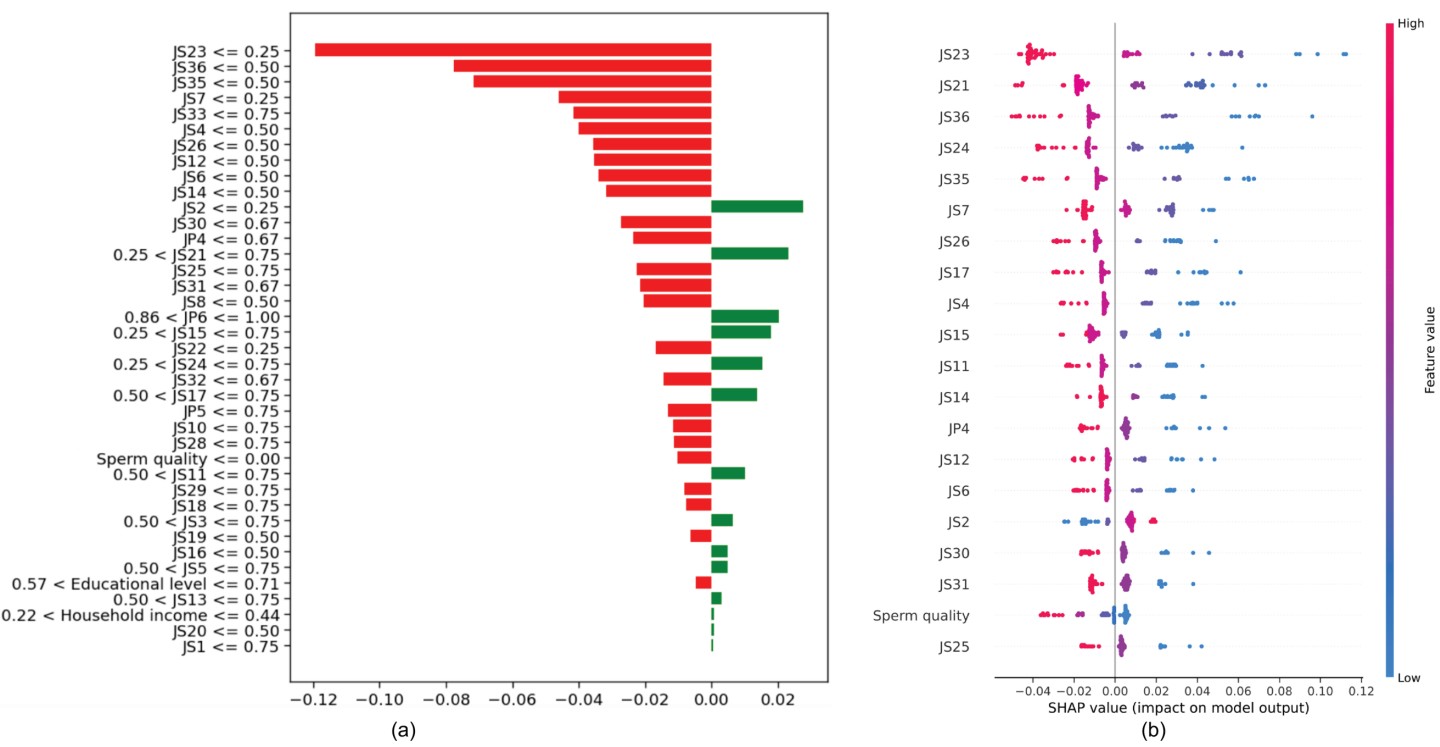

**Fig 17. Predictions for the stressed class of a randomly selected instance using (a) LIME and (b) SHAP. Here, the bars (LIME) and SHAP values indicate strong positive attributions for high-stress features such as workload (JS23) and unclear assignments (JS36), suggesting a higher probability of the stressed class.**

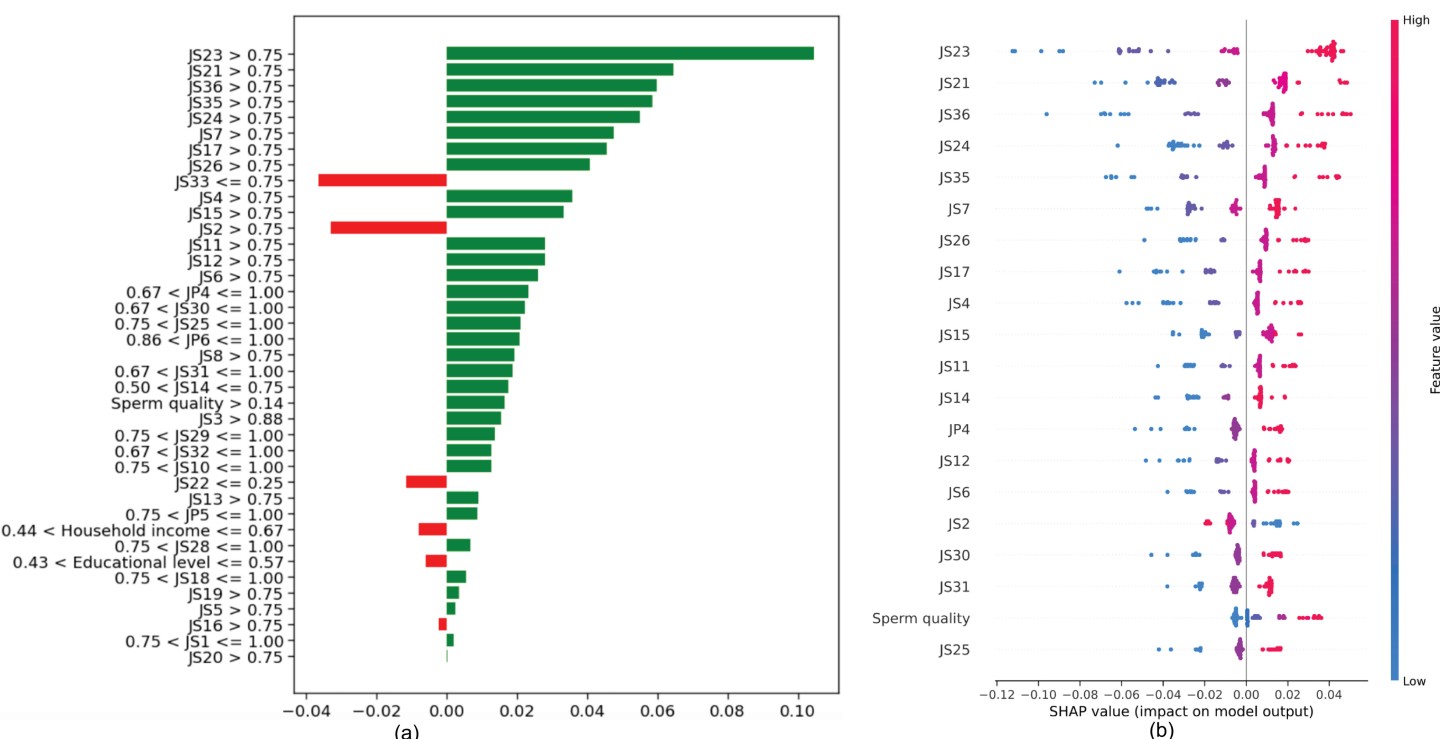

**Fig 18. Predictions for the not stressed class of a randomly selected instance using (a) LIME and (b) SHAP. In this low-stress example, features like positive colleague relationships (JS25) and good communication (JS33) have negative attributions (LIME) or negative SHAP values, pulling the prediction toward the not stressed class.**

bars in LIME and the negative SHAP values both indicate protective features that lower stress risk.

This bidirectional analysis of both global (Fig 15) and local (Figs 17 and 18) feature contributions provides actionable insights for workplace safety management. For instance, organizations should prioritize workload optimization (JS23) and clear task assignments (JS36) to reduce accident risks, while also fostering supportive communication (JS33) and recognition systems (JS17) as protective measures against stress-induced safety incidents. By understanding how these factors synergize or counterbalance at the individual level, safety managers can tailor interventions that effectively target stress mitigation and enhance overall workplace safety.

## Model deployment for workplace safety management

To facilitate practical implementation of stress-based safety monitoring, we deployed our model on the Hugging Face platform using Gradio (https://huggingface.co/spaces/JnS123456/Occupational_stress_detection). This deployed model provides the following benefits:

- Real-time stress level assessment for safety monitoring
- User-friendly interface for regular safety checks
- Immediate feedback for proactive safety intervention
- Integration capabilities with existing safety management systems

Fig 19 demonstrates the practical implementation of our model through the Hugging Face platform. The interface allows safety managers to input workplace stress indicators and receive immediate predictions. Fig 19a shows the model accurately identifying high-stress scenarios that require immediate safety intervention, while Fig 19b demonstrates the detection of normal stress levels indicating safe working conditions. This real-time assessment capability enables organizations to implement proactive stress monitoring as part of their comprehensive workplace safety programs, facilitating timely interventions before stress-related safety incidents can occur.

## Discussion

### Methodological significance

This study presents several significant advancements over existing research in occupational stress detection. First, our ensemble-based model integrating Random Forest, Logistic Regression, and Support Vector Classifier demonstrated superior predictive performance, achieving the highest accuracy (90.32%) and macro-averaged F1-score (89.20%) among evaluated methods. This performance notably exceeds previous state-of-the-art approaches that relied on single-model techniques or less comprehensive ensemble strategies.

Second, the robust hybrid feature selection approach, combining Recursive Feature Elimination with Cross-Validation (RFECV) and Analysis of Variance (ANOVA), significantly improved model performance by ensuring that both individually significant and interaction-sensitive features were included. Ablation studies confirmed the complementary nature of these methods, demonstrating that their integration captures critical occupational stress indicators overlooked by single-method approaches.

Third, leveraging natural language sentence generation from tabular survey data enabled the effective utilization of pre-trained large language models, notably BioBERT, which performed comparably to our best ensemble model. This data transformation approach facilitated deep contextual understanding, bridging the gap between traditional tabular analysis and advanced language models, thus expanding the methodological landscape for occupational stress detection.

Collectively, these contributions establish our method's superiority in predictive accuracy, interpretability, and generalizability, making it particularly suitable for real-world deployment in workplace safety management systems.

### Implications for workplace safety

The study has significant implications for various stakeholders in workplace safety management:

**Implications for Safety Management Systems:**

- **Real-time Monitoring:** The deployed model enables continuous stress monitoring as part of safety management systems, addressing the reactive nature of traditional approaches noted by [46].
- **Risk Assessment:** Integration of stress detection into safety protocols allows early identification of high-risk situations, particularly when multiple stress factors compound (e.g., high workload combined with unclear assignments).
- **Prevention Strategies:** The quantified impact of different stressors (e.g., 32% contribution from excessive workload) enables prioritized safety interventions.

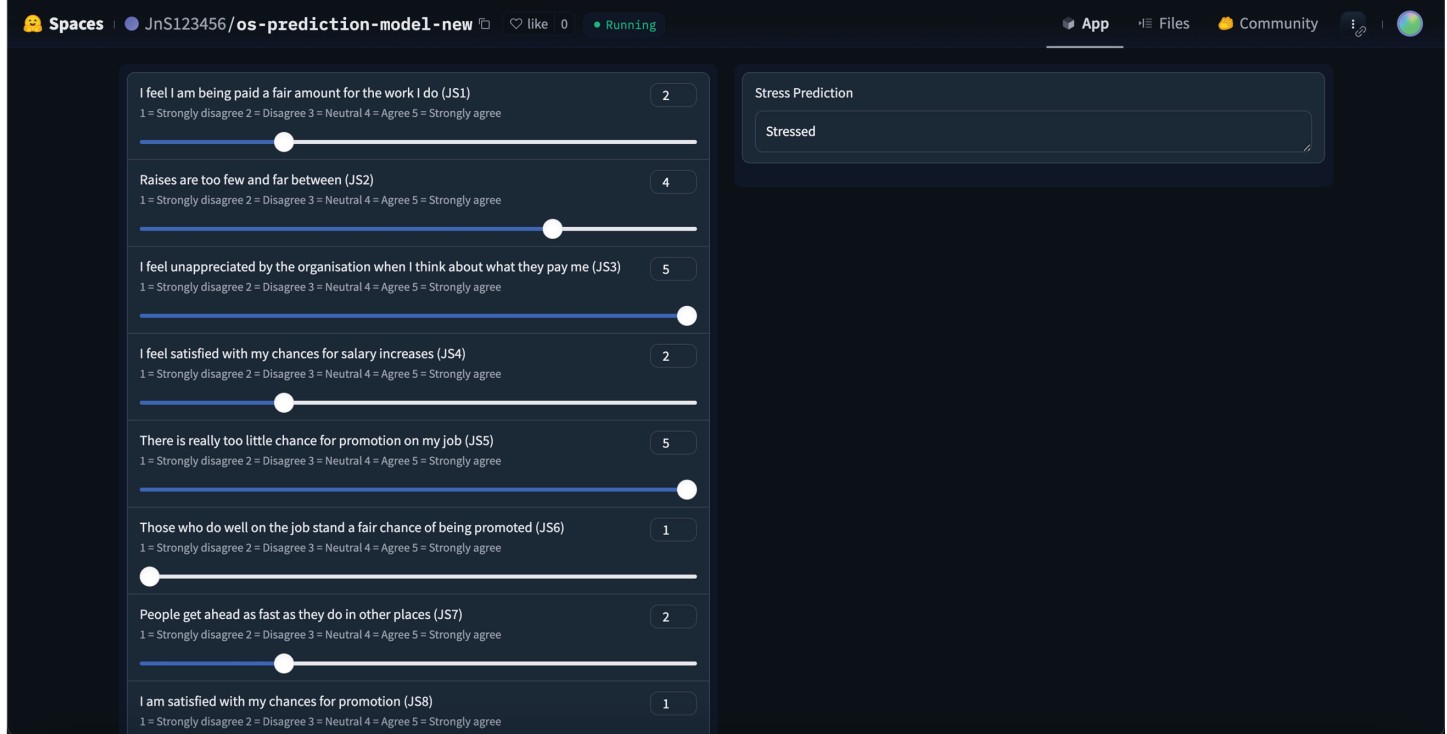

(a)

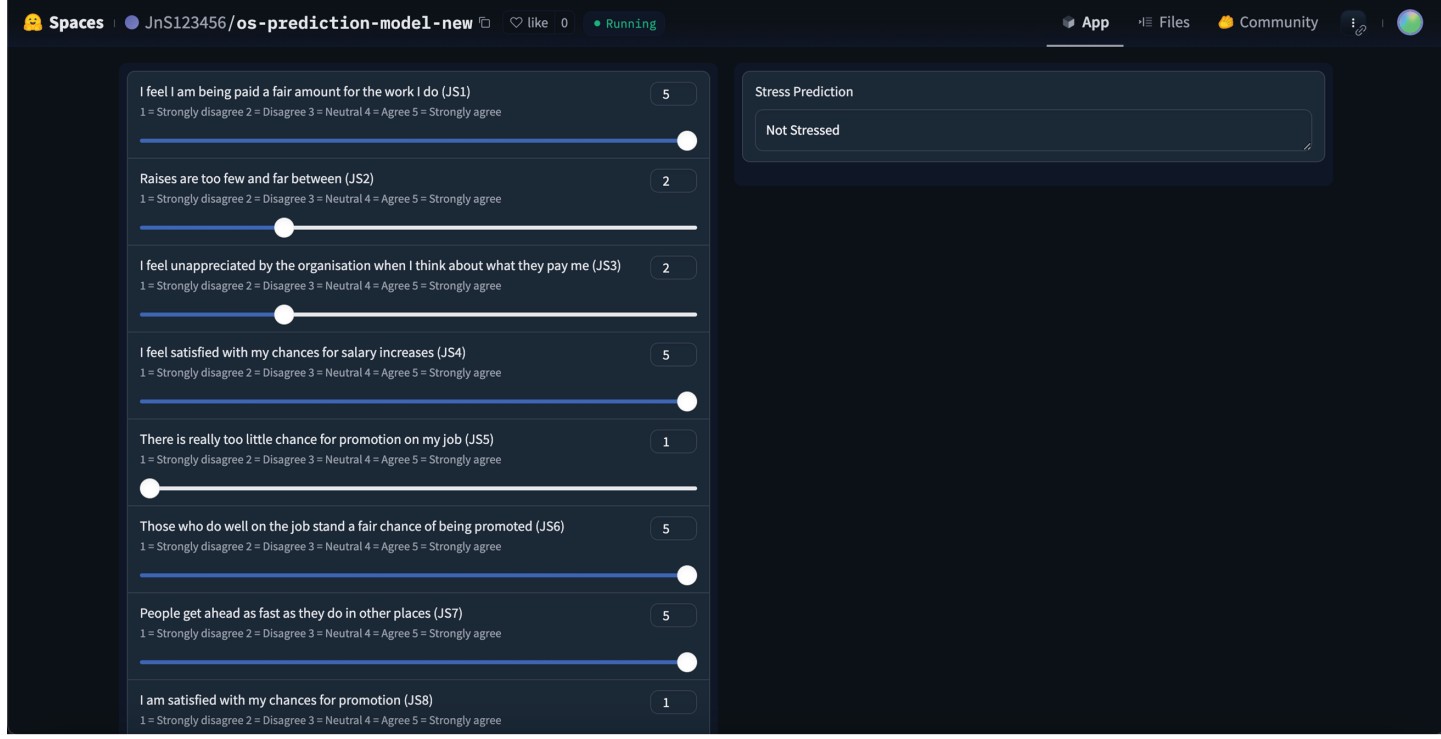

(b)

**Fig 19. Deployed model interface on Hugging Face platform showing stress detection capabilities for workplace safety monitoring. (a) Model prediction interface showing detection of high occupational stress, triggering safety intervention alerts. (b) Model prediction interface demonstrating detection of manageable stress levels, indicating safe working conditions.**

**Implications for safety professionals:**

- **Risk Evaluation:** Safety officers can use the model to assess how stress levels may compromise safety protocols and behavior.
- **Intervention Design:** The bidirectional analysis of stress factors (inducing vs. mitigating) enables targeted safety program development.
- **Performance Monitoring:** The tool provides objective metrics for evaluating the effectiveness of safety interventions.

**Implications for organizational safety culture:**

- **Policy Development:** Organizations can develop evidence-based safety policies that address both direct hazards and stress-related risks.
- **Training Programs:** The identified stress patterns can inform safety training programs that address both technical and psychosocial aspects.
- **Communication Strategies:** The importance of clear work assignments (stress reduction of 28%) suggests the need for improved safety communication protocols.

**Implications for regulatory framework:**

- **Standard Development:** The findings can inform occupational safety standards that incorporate stress management requirements.
- **Inspection Protocols:** Regulatory bodies can develop more comprehensive inspection protocols that include stress assessment.
- **Risk Classification:** The quantified stress impacts can help in developing risk classification systems for different workplace environments.

## Limitations and future work

While this study advances workplace safety through AI-driven stress detection, several limitations warrant attention:

- **Population Specificity:** The current model is based on Malaysian workplace data, potentially limiting its generalizability to different safety cultures and regulatory environments. Future research should improve generalizability by taking diverse populations into consideration.
- **Temporal Dynamics:** The cross-sectional nature of the data doesn't capture how stress patterns evolve over time in response to changing safety conditions. Future research may benefit by focusing on longitudinal studies examining the relationship between stress patterns and safety incidents.
- **Intervention Validation:** While the study identifies stress factors, the effectiveness of specific safety interventions based on these findings requires validation. Future research could design and test targeted interventions, evaluating their effectiveness using the proposed occupational stress detection and workplace safety models.

These future directions will further strengthen the connection between stress management and workplace safety, ultimately contributing to more effective occupational safety programs.

## Conclusions

This study presents a safety-centered, AI-driven framework for occupational stress detection that integrates machine learning, deep learning, and large language models into proactive workplace safety management. By combining a comprehensive preprocessing pipeline, multi-technique feature selection, and robust model development, we achieved a 90.32% accuracy, surpassing the performance of existing state-of-the-art methods. One crucial finding of the research is that combining RFECV and ANOVA techniques for feature selection yields better prediction accuracy than using them individually. Moreover, domain analysis using LLMs revealed that occupational stress is closely related to the biomedical domain than clinical or generalist domains, indicating that occupational stress rises from both physical and psychological factors rather than just one of them.

The employed methods in the study collectively offer a quantifiable and interpretable approach to understanding how organizational elements, particularly excessive workload and ambiguity (27%), poor communication (17%), and positive work environment (16%), impact occupational stress levels. Our three-fold validation techniques: holdout validation, cross-validation, and external validation with synthetic data, establishes the reliability and robustness of this framework, providing substantial evidence for its applicability in diverse settings.

Although our approach addresses critical gaps in current occupational stress research, there remain potential limitations related to population specificity, temporal dynamics, and intervention validation. Future work can extend this framework to different cultural contexts, employ longitudinal studies for capturing stress evolution, and integrate targeted interventions to validate effectiveness. By continually refining these components, organizations and safety practitioners can proactively mitigate the risks associated with occupational stress, ultimately fostering safer and more resilient work environments.

## Supporting information

**S1 Dataset. Synthetic data using CopulaGAN.**
(CSV)

**S2 Dataset. Synthetic data using CTGAN.**
(CSV)

**S3 Dataset. Synthetic data using GaussianCopula.**
(CSV)

**S4 Dataset. Synthetic data using TVAE.**
(CSV)

**S5 Text. Codebook for synthetic data.**
(PDF)

## Author contributions

**Conceptualization:** Mohammad Junayed Hasan, Jannat Sultana, Silvia Ahmed, Sifat Momen.

**Data curation:** Mohammad Junayed Hasan.

**Formal analysis:** Mohammad Junayed Hasan, Sifat Momen.

**Investigation:** Mohammad Junayed Hasan, Jannat Sultana, Silvia Ahmed, Sifat Momen.

**Methodology:** Mohammad Junayed Hasan, Jannat Sultana, Sifat Momen.

**Project administration:** Sifat Momen.

**Resources:** Mohammad Junayed Hasan, Silvia Ahmed.

**Software:** Mohammad Junayed Hasan.

**Supervision:** Silvia Ahmed, Sifat Momen.

**Validation:** Mohammad Junayed Hasan, Jannat Sultana, Sifat Momen.

**Visualization:** Mohammad Junayed Hasan, Jannat Sultana, Sifat Momen.

**Writing – original draft:** Mohammad Junayed Hasan.

**Writing – review & editing:** Mohammad Junayed Hasan, Jannat Sultana, Silvia Ahmed, Sifat Momen.

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
