## [Decision Letter · Decision Letter 0]

PONE-D-24-54716Early detection of occupational stress: Enhancing workplace safety with machine learning and large language modelsPLOS ONE

Dear Dr. Momen,

Thank you for submitting your manuscript to PLOS ONE. After careful consideration, we feel that it has merit but does not fully meet PLOS ONE’s publication criteria as it currently stands. Therefore, we invite you to submit a revised version of the manuscript that addresses the points raised during the review process.

We look forward to receiving your revised manuscript.

Kind regards,

Matthew Chin Heng Chua

Academic Editor

PLOS ONE

Additional Editor Comments (if provided):

Reviewers' comments:

Reviewer's Responses to Questions

**Comments to the Author**

1. Is the manuscript technically sound, and do the data support the conclusions?

Reviewer #1: Yes

Reviewer #2: No

Reviewer #3: Yes

Reviewer #4: Yes

Reviewer #5: Yes

2. Has the statistical analysis been performed appropriately and rigorously? 

Reviewer #1: Yes

Reviewer #2: I Don't Know

Reviewer #3: Yes

Reviewer #4: Yes

Reviewer #5: Yes

3. Have the authors made all data underlying the findings in their manuscript fully available?

Reviewer #1: Yes

Reviewer #2: No

Reviewer #3: Yes

Reviewer #4: Yes

Reviewer #5: Yes

4. Is the manuscript presented in an intelligible fashion and written in standard English?

Reviewer #1: Yes

Reviewer #2: Yes

Reviewer #3: Yes

Reviewer #4: Yes

Reviewer #5: Yes

5. Review Comments to the Author

Reviewer #1: The issues are listed in the following:

1. The professional English editing is recommended. The authors should get editing help from someone with full professional proficiency in English.

2. The abstract contains a large amount of information. It is recommended to simplify the language and highlight the innovative points and main results of the study. Some important methods suggested references is higher quality or a new literature, such as CNN "DOI: 10.1109 / TFUZZ. 2024.3369944", "DOI10.3389 / fpubh. 2022.981019"; Machine learning methods can refer to "DOI10.2174/1574893614666190416152025".

3. Although SHAP and LIME are used for model interpretation, the results primarily focus on feature importance, lacking in-depth analysis of the model's decision-making process.

4.  It is recommended to include an analysis of the differences between synthetic and real data, discuss the limitations of synthetic data generation methods, and propose improvements.

5. High-resolution images should be provided to ensure that all details are visible and can be thoroughly examined by readers.

6. It is recommended to expand the literature review section, particularly with a deeper discussion of recent advancements in AI technologies for occupational stress detection.

7. It is recommended to add a detailed description of the synthetic data generation methods in the methodology section, including the algorithms used, parameter settings, and quality assessment of the generated data.

8. What is the main difference or importance of the proposed methods and the other state-of-the-arts?

9. The conclusion should be concise and powerful, summarizing the main findings and contributions of the research.

10. The Conclusion section should point out the potential disadvantages and possible future research directions of the manuscript. How this work can be extended in future?

Reviewer #2: My comments are as follows:

1. How does the proposed AI-based framework combine machine learning, deep learning, and large language models to create a unified approach for occupational stress detection?

2. The manuscript mentions the integration of RFECV and ANOVA for feature selection. How were these techniques combined, and what criteria were used to determine the final set of 39 critical stressors?

3. Could you elaborate on the advanced preprocessing techniques used, particularly in managing imbalanced data or missing values?

4. The manuscript reports a 90.32% accuracy for the ensemble model and 89.00% on synthetic data. Were other performance metrics, such as precision, recall, F1-score, and AUC-ROC, evaluated to assess the model’s reliability in stress detection?

5. 3.It is always recommended to apply 10-fold CV for unbiased and reliable prediction results, which is not possible due to the random split of data during training and testing. It is kindly requested to consider these recommendations and add results OR discussion in the revised version with reference to the suggested references provided.

a. https://doi.org/10.1016/j.conbuildmat.2019.07.224

b. https://doi.org/10.1080/27684830.2023.2201015

6. The use of Explainable AI techniques is highlighted in the study. How were these techniques implemented, and what specific visualizations or interpretability tools were used to quantify the impact of workplace safety factors?

7. The study identifies excessive workload, unclear work assignments, and poor organizational communication as primary stressors. How were these percentages (e.g., 32%, 28%) quantified, and what statistical or machine learning methods supported these findings?

Reviewer #3: Authors investigated an AI-based framework for proactive occupational stress detection to improve workplace safety. However, before acceptance, major corrections are required:

1.In abstract, please clearly specify the Purpose, Contribution, and findings.

2.Please write down the organization of the paper written end of Introduction.

3.Please provide a mathematical model of proposed 1D CNN.

4.Please analysis computational complexity in your proposed model.

5.It is recommended that professional proofreaders and native English corrections be used.

Reviewer #4: The authors have put sound efforts into addressing the research gap of the early detection of occupational stress in women. This paper explores the application of 11 machine learning techniques, 1D-CNN and five LLM in detecting stress among women.

The work is acceptable subject to the following minor modifications:

1.It claimed that a framework has been developed with ability to process diverse data types and provide explainable results, however, no such framework is presented in the manuscript.

2.In Figure 1, the alphabetical numbering of the captions needs to be corrected.

3.Why in Figure 2, Box plots show a same range for the data distribution for six job performance survey questions?

4.One of the contributions is empirical demonstration by extracting more predictive features than individual methods, detail explanation is missing.

5.More explanation of figure 12 and 13 is required to be added.

6.Figure 14 visibility should be increased.

Reviewer #5: PAper is well written and significantly addressed the research problem successfully by employment machine learning models and statistically analysed results. All results well tabulated with grpahs. Also model has been generalised too

6. PLOS authors have the option to publish the peer review history of their article (what does this mean?). If published, this will include your full peer review and any attached files.

Reviewer #1: No

Reviewer #2: No

Reviewer #3: **Yes: **Saifur Rahman Sabuj

Reviewer #4: No

Reviewer #5: No

---

## [Author Response · Author response to Decision Letter 1]

25 Mar 2025

Responses to the reviewer and editor comments have been provided in a file titled "Response to Reviewers". Please find the responses in the relevant file. Thank you.

---

## [Decision Letter · Decision Letter 1]

Early detection of occupational stress: Enhancing workplace safety with machine learning and large language models

PONE-D-24-54716R1

Dear Dr. Momen,

We’re pleased to inform you that your manuscript has been judged scientifically suitable for publication and will be formally accepted for publication once it meets all outstanding technical requirements.

Kind regards,

Ashad Kabir, PhD

Academic Editor

PLOS ONE

Additional Editor Comments (optional):

The authors have adequately addressed the reviewers' comments, and the reviewers have recommended the manuscript for acceptance.

Reviewers' comments:

Reviewer's Responses to Questions

**Comments to the Author**

1. If the authors have adequately addressed your comments raised in a previous round of review and you feel that this manuscript is now acceptable for publication, you may indicate that here to bypass the “Comments to the Author” section, enter your conflict of interest statement in the “Confidential to Editor” section, and submit your "Accept" recommendation.

Reviewer #1: All comments have been addressed

Reviewer #3: All comments have been addressed

Reviewer #4: All comments have been addressed

Reviewer #5: (No Response)

2. Is the manuscript technically sound, and do the data support the conclusions?

Reviewer #1: Yes

Reviewer #3: Yes

Reviewer #4: Yes

Reviewer #5: (No Response)

3. Has the statistical analysis been performed appropriately and rigorously? 

Reviewer #1: Yes

Reviewer #3: Yes

Reviewer #4: Yes

Reviewer #5: (No Response)

4. Have the authors made all data underlying the findings in their manuscript fully available?

Reviewer #1: Yes

Reviewer #3: Yes

Reviewer #4: Yes

Reviewer #5: (No Response)

5. Is the manuscript presented in an intelligible fashion and written in standard English?

Reviewer #1: Yes

Reviewer #3: Yes

Reviewer #4: Yes

Reviewer #5: No

6. Review Comments to the Author

Reviewer #1: The author has answered all my questions, and in; Some modifications have been made in the paper and it is recommended that the paper be published.

Reviewer #3: Thanks for correction. Please update the related work section. Some important paper in 2024 and 2025 are missing.

1) An end-to-end lightweight multi-scale CNN for the classification of lung and colon cancer with XAI integration

Reviewer #4: (No Response)

Reviewer #5: All Reviewer comments addressed and paper be accepted. The paper be accepted without any further revision.

7. PLOS authors have the option to publish the peer review history of their article (what does this mean?). If published, this will include your full peer review and any attached files.

Reviewer #1: No

Reviewer #3: No

Reviewer #4: **Yes: **Prof. Sania Bhatti

Reviewer #5: No

---

## [Editor Report · Acceptance letter]

PONE-D-24-54716R1

PLOS ONE

Dear Dr. Momen,

I'm pleased to inform you that your manuscript has been deemed suitable for publication in PLOS ONE. Congratulations! Your manuscript is now being handed over to our production team.

Kind regards,

on behalf of

Professor Ashad Kabir

Academic Editor

PLOS ONE